# Antidiabetic activities of *Bolanthus spergulifolius* (Caryophyllaceae) extracts on insulin-resistant 3T3-L1 adipocytes

**Gizem Ece Derici**[1], **Sibel Özdaş**[1]*, **İpek Canatar**[1], **Murat Koç**[2]

**1** Faculty of Engineering Sciences, Department of Bioengineering, Adana Alpaslan Türkeş Science and Technology University, Adana, Turkey, **2** Graduate School of Public Health, Department of Traditional, Complementary and Integrative Medicine, Ankara Yıldırım Beyazıt University, Ankara, Turkey

* sozdas@atu.edu.tr

**Data Availability Statement:** All data files are available from the Zenodo database with DOI number: 10.5281/zenodo.4824591 or URL link: http://doi.org/10.5281/zenodo.4824591.

## Abstract

Diabetes mellitus (DM) is a metabolic disorder with chronic hyperglycemia featured by metabolic outcomes owing to insufficient insulin secretion and/or insulin effect defect. It is critical to investigate new therapeutic approaches for T2DM and alternative, natural agents that target molecules in potential signal pathways. Medicinal plants are significant resources in the research of alternative new drug active ingredients. *Bolanthus spergulifolius (B. spergulifolius)* is one of the genera of the family Caryophyllaceae. In this study, it was explored the potential anti-diabetic effects *in vitro* of *B. spergulifolius* extracts on 3T3-L1 adipocytes. The total phenolic contents (TPC) of methanolic (MeOH), ethyl acettate (EA) and aqueous extracts of *B. spergulifolius* were evaluated via Folin-Ciocateu. *B. spergulifolius* extracts showing highly TPC (Aqueous< MeOH< EA) and their different concentrations were carried out on preadipocytes differentiated in to mature 3T3-L1 adipocytes to investigate their half-maximal (50%) inhibitory concentration ($IC_{50}$) value by using Thiazolyl blue tetrazolium bromide (MTT) assay. The $IC_{50}$ of MeOH, EA and Aqueous extracts were observed as 305.7 ± 5.583 μg/mL, 567.4 ± 3.008 μg/mL, and 418.3 ± 4.390 μg/mL and used for further experiments. A live/dead assay further confirmed the cytotoxic effects of MeOH, EA and Aqueous extracts (respectively, 69.75 ± 1.70%, 61.75 ± 1.70%, 70 ± 4.24%, and for all p< 0.05). Also, effects of extracts on lipid accumulation in mature 3T3-L1 adipocytes were evaluated by Oil-Red O staining assay. The extracts effectively decreased lipid-accumulation compared to untreated adipocytes (for all p< 0.05). Moreover, effect of extracts on apoptosis regulated by the *Bax* and *Bcl-2* was investigated by quantitative reverse transcription polymerase chain reaction (qRT-PCR). The extracts significantly induced apoptosis by up-regulating pro-apoptotic *Bax* expression but down-regulated anti-apoptotic *Bcl-2* gene expression compared to untreated adipocytes (for all p< 0.05). The *Glut-4* expression linked with insulin resistance was determined by qRT-PCR, Western-blot analysis, and immunofluorescence staining. In parallel, the expression of Glut-4 in adipocytes treated with extracts was significantly higher compared to untreated adipocytes (for all p< 0.05). Extracts significantly suppressed cell migration after 30 h of wounding in a scratch-assay (for all p< 0.05). Cell morphology and diameter were further evaluated by phase-contrast microscopy, scanning electron microscopy, Immunofluorescence with F-Actin and Giemsa staining. The adipocytes treated with

**Funding:** The author(s) received funding for this work by Adana Science and Technology University Scientific Research Projects Coordination Unit (Award Number: Project No: 19332006 and Project No: 20103005 | Recipient: Sibel Özdaş, PhD). The funder had no role in study design, data collection and analysis, decision to publish, or preparation of the manuscript.

**Competing interests:** The authors have declared that no competing interests exist.

extracts partially lost spherical morphology and showed smaller cell-diameter compared to untreated adipocytes (for all p< 0.05). In conclusion, these results suggest that extracts of *B. spergulifolius* cause to an induce apoptosis, decrease lipid-accumulation, wound healing, up-regulating Glut-4 level and might contribute to reducing of insulin-resistance in DM.

## Introduction

Diabetes mellitus (DM) is a metabolic disorder with chronic hyperglycemia featured by metabolic outcomes owing to insufficient insulin secretion and/or insulin effect defect. Chronic hyperglycemia causes cardiovascular diseases in DM and various complications on the eye, kidney and nervous system [1]. DM is etiologically classified into three main groups; type 1 DM (T1DM), type 2 DM (T2DM) and gestational DM (GDM). T2DM is the most common type of diabetes among the patients [2]. In T2DM, there is a progressive insulin release defect on the basis of insulin resistance, and insulin insufficiency [3].

Global DM prevalence was reported as 425 million in 2017, 463 million in 2019, and 578 million in 2030 and it is estimated to boost to 700 million in 2045 [1, 2]. The dramatic increase in DM is largely due to T2DM and metabolic and endocrine dysfunction involving adipose tissue. Metabolic-dysfunction, changes in endocrine secretions and chronic inflammation of adipose tissue conduce to improving insulin resistance both locally and systemically [4]. Therefore, it is critical to manage T2DM to improve public health by controlling DM, reducing systemic complications and improving insulin sensitivity of adipose tissue [5]. Moreover, developing drug-resistance mechanisms, toxic effects, and high-cost for current drugs in therapy are still major problems for T2DM [6]. Therefore, it is very important to investigate new therapeutic approaches for DM and alternative anti-diabetic agents that target molecules in potential signal pathways, which are natural, accessible, cost-effective, and have low toxicity. Medicinal plants are very significant resources in the research of alternative new drug active ingredients [7].

Caryophyllaceae species are rich in phytochemicals and their major chemical components are phenolic substances [8]. Caryophyllaceae is one of the largest families that includes approximately 100 genera and 3000 species [9, 10]. The family Caryophyllaceae found commonly in the Northern Hemisphere, especially in the Mediterranean region, rarely in the Southern Hemisphere [9, 11].

The family Cayophyllaceae have antiviral, antifungal, immunomodulatory, antidiabetic, antiinflammatory, antioxidant and anticancer effects and their ethnomedicinal uses is known [8]. It has been determined that Caryophyllaceae crude plant extracts can be an important pharmacological resource in developing potential agents against diabetes in addition to high antioxidant capacity and various bioactive properties [12]. New approaches to the prevention/treatment of hyperglycemia have popularized the therapeutic uses of α-amylase and α-glucosidase inhibitors. It has also been reported that Caryophyllaceae species have significant α-glucosidase inhibitory action and are a potential natural source of α-glucosidase enzyme inhibitors in the treatment of DM [13]. Moreover, phenolic compounds in plants have inhibition activities on α-glucosidase/amylase enzyme cause to decrease blood glucose level [3, 13].

*Bolanthus* (Ser.) Reichb., is one of the genera of the family Caryophyllaceae. The aspect that makes this genus important for Turkey is that the entire taxa are endemic to Turkey [11, 14, 15]. *Bolanthus spergulifolius* (Jaub. & Spach) Hub.-Mor. (*B. spergulifolius)* was first defined with the name *Heterochroa spergulifolia* by Jaubert and Spach in 1843 [16]. Later on, the taxon

was transferred to the genus *Bolanthus* by Huber-Morath in 1967 [15]. Cayophyllaceae family has been traditionally used in therapeutic medicine for some deseases [8]. Some studies have shown that Cayophyllaceae family has compounds such as saponins, flavonoids, phenolic acids, phytoecdysteroids with strong bioactive properties and can be resources in the research of new drug active ingredients [12]. However, biological activities of *B. spergulifolius* are poorly described yet. Therefore, in this study, it was aimed to investigate the anti-diabetic effects *in vitro* of *B. spergulifolius* extracts on 3T3-L1 adipocytes.

## Material and methods

### Chemicals and reagents

Dimethyl sulfoxide (DMSO), Bovine serum albumin (BSA), Dexamethasone (DEX), Insulin, Glutaraldehyde, Ethylacetate (EA), Methanol (MeOH), Ethanol, +/+ PBS (containing Ca++ and Mg++), Phosphate buffered saline (PBS), İsobutylmethylxanthine (IBMX), Formalin, Isopropanol, 2-propanol, 4'6-diamidino-2-phenylindole (DAPI), Cloroform TRIzol, DEPC water, Lipid (Oil Red O) Staining Kit, Glycine, Sodium dodesylsulfate, Tris, 3-(4,5-dimethylthiazol-2-yl)-2,5-diphenyltetrazolium bromide (MTT), Mercaptoethanol, RIPA buffer, Protease inhibitor cocktail, Skim milk powder, Folin-Ciocalteu, Sodium carbonate, Gallic acid, Triton X-100, Tween 20, Paraformaldehyde, Hexamethyldisilazane (HMDS), Giemsa were optained from Sigma-Aldrich (St Louis, MO, USA). Dulbecco's Modified Eagle Medium (DMEM), L-Glutamine, Fetal bovine serum (FBS), Penicillin, Streptomycin were obtained from Hyclone (GE Healthcare, USA). Quick Start Bradford Protein Assay Kit 1, Immun-Blot PVDF membrane, 2x Laemmli sample buffer, 10x Tris/Glycine/SDS buffer, ECL reagent were purchased from Bio-Rad (Hercules, USA). Calcein-Acetometomethoxy (C-AM) (ab141420), Ethidium homodimer-1 (Eth-1) (ab145323), anti-Glut-4 (ab35826 and ab654) anti-β-Actin (ab8226), Goat Anti-Mouse IgG secondary-antibody (ab205719), Goat Anti-Mouse IgG H&L (Alexa Fluor® 488) (ab150113) Phalloidin-iFluor 488 Reagent (ab176753) and BrightMount/Plus were purchased from Abcam (Cambrige, UK).

**Plant material.** *B. spergulifolius* used in this study was collected from Kütahya province, Gediz district, Murat Mountain, 1495 m, Kaplıcalar road, 38˚56'48" N—29˚36'44" E, serpentine soils, grassy levels in Turkey during the flowering season in 2019. Identification and authentication of the plant material were conducted by Dr. Murat Koç, taxonomist, from Department of Traditional, Complementary and Integrative Medicine Graduate School of Public Health, Ankara Yıldırım Beyazıt University, Turkey. A voucher specimen of *B. spergulifolius* (No. Koç2043&Hamzaoğlu) were deposited in the herbarium (Biology Department of Bozok University, Turkey) [11]. The air-dried plant materials under shade were ground in a coffee grinder (Krups 75, France) [17].

### Preparation of *B. spergulifolius* extracts

For *B. spergulifolius* MeOH extraction; 5 g of dried plant samples was extracted with 100 mL MeOH for 1 h in an ultrasonic bath (Ultracleaner, DH-WUC-06H, Daihan, USA) at 15˚C. These poses were replicated until the solvent turned colorless. After the plant extract was filtered using Whatman No. 1. Then the MeOH was removed from the filtrate of extract by a rotary evaporator (Buchi Rotavapour R-200, Marshall Scientific, Hampton, USA) at 37˚C. Then, the filtrate was stored at -80˚C.

For *B. spergulifolius* EA extraction; 5 g of dried plant samples was extracted with 100 mL EA and for 1 h in an ultrasonic bath (Ultracleaner, DH-WUC-06H, Daihan, USA) at 15˚C (10 min intervals in three-times). After the filtration of extracts were performed using Whatman No. 1. The filtrate of extract was concentrated by a rotary evaporator (Buchi Rotavapour R-200, Marshall Scientific, Hampton, USA) at 37˚C. Then, the filtrate was stored at -80˚C.

For *B. spergulifolius* Aqueous extraction; 5 g of dried plant samples was extracted with 100 mL water for 1 h in an ultrasonic bath (Ultracleaner, DH-WUC-06H, Daihan, USA) at room temperature (10 min intervals in triplicates) and filtered (Whatman No.1) under vacuum. The filtrate of Aquaeous was frozen at -20°C and then dried with freeze dryer (BK-FD10P, Biobase, China).

The extracts of plant samples were weighed (TP-214 Timberline, Denver Instrument Company, UK) and the yield of all of the extraction was calculated in terms of (w/w) % per g of crude extract. The extracts were reconstituted in DMSO at a final-concentration and stored at -20°C under shade until further studies.

## Folin-Ciocalteu

Total phenolic content (TPC) was evaluated by the Folin-Ciocalteu method [18]. 100 µL 10% (v/v) of Folin-Ciocalteu reagent was added to 20 µL samples of each plant extract for 5 min at room temperature. Finally, 80 µL 2% (w/v) of sodium carbonate were added and incubated in the dark for 30 min at 40°C. Then the absorbance was measured at wavelength of 765 nm with a spectrophotometer (Uvmini-1240, Shimadzu, Kyoto, Japan). According to the standard calibration curve, value of TPC for each extract was presented as gallic acid equivalents (mg GAE/ g of dry weight).

## Cell culture and differentiation of 3T3-L1 into mature adipocytes

3T3-L1 (CRL-173) preadipocytes were optained from the American Type Culture Collection (ATCC, Manassas, VA). Preadipocyte expansion medium was prepared 90% DMEM supplemented with 10% FBS, 1% L-Glutamine and 1% Penicillin+streptomycin antibiotics. The cells were cultured in 25 cm$^2$ flask containing a preadipocyte expansion medium. The cells were grown in an incubator (Nuaire, NU-5800, USA) with a humidified air at 37°C containing 5% CO$^2$. The culture medium was altered every two-three day in cabinet (LabGard NU-540, Nuaire, US) and cells observed using an inverted microscope (Leica, Dmil Led Fluo, Thermo Fisher, Germany).

The differentiation of 3T3-L1 preadipocytes into mature adipocytes were cultured and maintained [19]. Differentiation medium was prepared 90% DMEM supplemented with 10% FBS, 1% L-Glutamine, 1% Penicillin+streptomycin antibiotics-containing DEX (1µM), Insulin (1 µg/mL), and IBMX (0.5 mM). First, 3T3-L1 cells were seeded preadipocyte expansion medium at 8x10$^4$ in 6-well plate containing 1 mL medium until reached confluence. After 48 h for the induction of differentiation 3T3-L1 preadipocyte were cultured with differentiation medium. After 72 h the induction of differentiation, the differentiation medium was removed and added by adipocyte maintenance DMEM supplemented with 10% FBS and insulin (1µg/mL) for another 48 h. After 5 days the induction of differentiation, the fresh adipocyte maintenance medium was replaced every two days until 14 days. The visible of lipid accumulation was monitored with an inverted microscope (Leica, Dmil Led Fluo, Thermo Fisher, Germany) [20, 21]. The mature 3T3-L1 adipocytes were used for further analysis.

## Oil Red O staining assay

To visualize and measure of the lipid accumulation in preadipocytes, mature 3T3-L1 adipocytes and the effects of *B. spergulifolius* extracts on adipogenesis in 3T3-L1 adipocytes, was performed Oil Red O Staining Kit following the manufacturer's instructions [20, 21]. The 3T3-L1 preadipocytes and adipocytes were cultured into a 24-well plate (2x10$^4$ cells/well). After confluence, the medium was removed and cells were washed twice with PBS and fixed with 100 µL 10% (v/v) of formalin for 60 min at room temperature. The fixed cells were washed twice with PBS and then added with 100 µL 60% (v/v) of isopropanol for 5 min at room temperature. Oil

Red O stock solution was prepared with 100% (v/v) isopropanol and was kept in 56˚C water bath for 1 h. The Oil Red O working solution was prepared with 3-unit of Oil Red O stock solution to 2-unit water, and shook (Everlast Rocker 247, Benchmark, USA) for 10 min and filtered using Whatman no:1. The cells were incubated for 20 min at room temperature with working solution 100 μL. Then, the working solution was removed and wells were washed three times with water. The wells were added Hematoxylin for 1 min and then washed three times with water. Finally, cells were coated with water and images of them were taken under a microscope. Also, lipid accumulation in cells was quantified by eluting Oil Red O stain using isopropanol and optical density (OD) was measured at a wavelength of 490 nm using a spectrophotometer (Nanoprop 2000c, Thermo Scientific, USA) [22].

## Thiazolyl blue tetrazolium bromide (MTT) assay

MTT was used to determine the effect of *B. spergulifolius* extracts on 3T3-L1 cell viability and the dosage of the extracts for further studies [23]. 3T3-L1 adipocytes were cultured in a 96-well plate ($2x10^3$ cells/well) and incubated overnight at 37˚C under 5% $CO^2$. After 24 h, the medium was removed and 3T3-L1 adipocytes were treated with fresh medium containing different concentrations of plant extracts (for MeOH extract, 0–1000 μg/mL; for EA extract; 0–2000 μg/mL; for Aqueous extract, 0–1000 μg/mL) or 0.1% (v/v) DMSO for 24 h at 37˚C. Then, the medium was removed and cells were washed twice with PBS. MTT solution (0.5 g/mL in PBS) was supplemented in each well at 100 μL and the cells were incubated for 4 h at 37˚C. After MTT solution was suspended and 100 μL of DMSO was supplemented to each well. 96-well plates wrapped with aluminum foil and placed in a shaker (Everlast Rocker 247, Benchmark, USA) for about 1 h. The color change observed in the wells and the relative cell viability were calculated by determining with a spectrophotometer (Uvmini-1240, Shimadzu, Kyoto, Japan) absorbance at 570 nm. The results of cytotoxicity were presented as the relative reduction of the $OD_{570}$, which correlated with the number of viable cells in relation to the percentage of control cells (as taken 100%) using the following formula. The cells half-proliferation inhibitory concentration ($IC_{50}$) for MeOH, EA, Aqueous extracts of *B. spergulifolius* were calculated from the linear regression equation. The relative cell-viability and $IC_{50}$ were plotted in graphs.

$$\text{Cell viability } (\%) = (\text{Average } OD_{570} \text{ of treated cells}/\text{Average } OD_{570} \text{ of control cells}) \text{ x } 100 \quad (1)$$

## Live-dead staining assay

Live-dead staining was conducted to evaluate the effect of *B. spergulifolius* extracts on 3T3-L1 cell viability [24]. C-AM and Eth-1 dyes were used after 24 h treated with plant extracts. First, cells incubated in 24 well-plate ($2x10^4$ cells/well) were removed medium and washed with 250 μL +/+ PBS twice. 1 μM C-AM and 1 μM Eth-1 dissolved in 250 μL +/+ PBS was added to each well and cells were incubated for 30 min in the dark at room temperature. After that, cells were removed the solution, and washed with 250 μL +/+ PBS. The cells were visualized at 10X under confocal microscope confocal microscope (Carl-Zeiss LSM 710 NLO, GmbH, Germany) to evaluate the spatial dispersion of the living (green/C-AM) and dead (red/Eth-1) cells. The ImageJ software (imagej.nih.gov) was used for quantification of relative cell-viability in each field [24]. The viable cells (%) were calculated:

$$\text{Viable Cells } (\%) = [\text{Viable cells}/(\text{Viable cells} + \text{Dead cells})] \times 100 \quad (2)$$

## Quantitative reverse transcription polymerase chain reaction (qRT-PCR)

To investigate the effect of *B. spergulifolius* on apoptosis and insulin resistance, respectively the expression of *Bcl-2-associated X (Bax)*, *B-cell lymphoma-2 (Bcl-2)* and *Glucose transporter-4 (Glut-*

**Table 1. Genes and sets of the primers used in qRT-PCR.**

| Gene | Gene Bank Accession ID | Primer forward 5'-3' nucleotide sequences | Primer reverse 5'-3' nucleotide sequences | Product length (bp) |
|---|---|---|---|---|
| [a]*Bax* | NM_007527.3 | CCAGGATGCGTCCACCAAGA | GGTGAGGACTCCAGCCACAA | 443 |
| [b]*Bcl-2* | NM_009741.5 | TGAGTACCTGAACCGGCATCT | GCATCCCAGCCTCCGTTAT | 57 |
| [c]*Glut-4* | NM_001359114.1 | CTTCTTTGAGATTGG CCCTGG | AGGTGAAGATGAAGAAGCCAAGC | 215 |
| [d]*Gapdh* | NM_001289726.1 | CAAGGTCATCCATGACAACTTTG | GTCCACCACCCTGTTGCTGTAG | 690 |

[a]*Bax*: *Bcl-2-associated X*

[b]*Bcl-2*: B-cell lymphoma 2

[c]*Glut-4*: Glucose Transporter 4

[d]*Gapdh*: Glyceraldehyde-3-Phosphate Dehydrogenase

*4)* genes at the mRNA level in 3T3-L1 adipocytes following treatment of extracts was evaluated using quantitative qRT-PCR [25, 26]. The 3T3-L1 adipocytes were seeded in 6-well plates at an initial density of $3x10^5$ cells/well. Total RNA was isolated from the cells using TRIzol reagent 24 h of treatment with plant extracts according to the product manufacturer's protocol. The RNA was dissolved in DEPC water and stored at -80°C until it was used. The concentration of RNA samples was measured using a spectrophotometer (Nanoprop 2000c, Thermo Scientific, USA). The 1 μg of the total RNA with 260/280 nm ratio: 1.8–2.1 were used for complementary DNA (cDNA) synthesis. ProtoScript II first Strand cDNA Synthesis Kit (NEB, New England Biolabs Inc., USA) was used to convert the RNA to cDNA following instructions of the manufacturer. The amplification of cDNA was prepared using the One Taq 2X Master Mix with the Standard Buffer (New England Bio labs Inc., USA) in a final volume of 25 μL using thermal cycler (SureCycler 8800, Agilent Technologies, USA) using primers synthesized by Sentebiolab, Turkey (Table 1). The qRT-PCR was performed using the GoTaq® 2-Step RT-qPCR System (Promega, ABD) on the Qiagen Rotor-Geneq (Qiagen, Hilden, Germany) by following PCR steps summarized in Table 2. The expression of target genes was analysed with the comparative threshold cycle $C_T$ method ($^{\Delta\Delta}C_T$) method, by which the results of each gene expression were normalized with the expression of *Glyceraldehyde-3-Phosphate Dehydrogenase (Gapdh)* gene [27]. The results of qRT-PCR were analysed with Rotor Gene Q Software 1.2 program (Qiagen, Germay).

## Western-blot

The method was performed to evaluate the effect of *B. spergulifolius* extracts on level of Glut-4 protein in 3T3-L1 adipocytes. The 3T3-L1 adipocytes were cultured in 6-well plate ($3x10^5$ cells/well). After treatment of cell with extract for 24 h, the medium was removed and each well was washed with 1 mL cold PBS twice. Then 225 μL RIPA buffer containing 10% Protease inhibitor cocktail was added to each well on cold block and the plate was rocked orbital shaker

**Table 2. The protocol PCR.**

| PCR step | Temperature (°C) | Time (min) | Cycle |
|---|---|---|---|
| İnitial denaturation | 94 | 3 | 1 |
| Denaturation | 94 | 0.5 | 35 |
| Annealing | 52–58 | 0.5 | 35 |
| Extension | 72 | 1 | 35 |
| Final extension | 72 | 5 | 1 |

PCR: Polymerase chain reaction

(Everlast Rocker 247, Benchmark, USA) for 15 min at +4˚C. The cells were scrapped, exposed with (Bioruptor sonicator, Belgium) and centrifuged at 15.000 rpm for 10 min. The protein pellet concentration was measured using Quick Start Bradford Protein Assay Kit 1 (Bio-Rad, Hercules, USA).

Protein samples (20 μg) were treated with Mercaptoethanol containing 5% 2x Laemmli sample buffer and loaded to 10% SDS-PAGE gel. Each sample was ran with 10x Tris/Glycine/ SDS buffer for 1 h at 100 V using Mini Pretean Tetra Cell Module (Bio-Rad, Hercules, USA) and transferred onto Immun-Blot PVDF membrane using Mini Trans Blot Cell Module (Bio-Rad, Hercules, USA). The membranes were blocked in 5% skim milk powder and incubated with anti-Glut-4 and anti-β-Actin as an internal control [for two, 1:100 (v/v) in 5% skim milk powder] at overnight at 4˚C. Membranes were then treated with Goat Anti-mouse IgG secondary antibody [1:1000 (v/v) 5% skim milk powder] for 2 h at room temperature and washed. Protein band was visualized with 1mL ECL reagent for 5 min in dark using VersaDoc (4000MP, Bio-Rad, Hercules, USA) and ImageLab software (Bio-Rad, Hercules, USA) was used for protein quantification [28].

## Immunofluorescence staining

Immunofluorescence staining was performed to determine the effect of *B. spergulifolius* extracts on intracellular level of Glut-4 in 3T3-L1 adipocytes. The cells were cultured on coverslips (Nunc Thermanox Coverslips, Thermo Scientific, USA) into a 6-well plate ($3x10^5$ cells/well) and then confluent cells treated with plant extracts. After 24 h, coverslips were transferred into 24-well plate and cells were washed twice with PBS. Then cells were fixed with 250 μL 4%lük (w/v) of paraformaldehyde in PBS for 10 min at room temperature. Fixed cells were wash with PBS and permeablized with 250 μL 0.1% (v/v) Triton X-100 in PBS for 10 min at room temperature. After permeabilization, cells were added to block nonspecific binding 250 μL 1% (w/v) of BSA in [PBST (PBS + %0.1 Tween 20)] for 10 min at room temperature. Blocked cells were incubated with 250 μL 1:100 (v/v) of anti-Glut4 and anti-Beta Actin in 1% (w/v) of PBST overnight at 4˚C in a humidified room. Then cells were incubated with 250 μL 1:1000 (v/v) of Goat Anti-Mouse IgG H&L (Alexa Fluor® 488) as secondary antibody for 60 min at room temperature in the dark. The nucleus was stained with 250 μL 1/1000 (v/v) of DAPI (1 μg/mL) in BSA for 5 min at room temperature. Finally, coverslips were coated with 3 μL BrightMount/Plus. The staining was imaged at 10X using a fluorescence microscope (Leica, Dmil Led Fluo, Thermo Fisher, Germany) and fluorescein signals were analysed ImageJ software (imagej.nih.gov) [29, 30].

Also, intra-ctyoplasmic staining was performed to examine the effect of each extract on cytoskeleton of 3T3-L1 adipocytes. The medium was discarded and washed with PBS. Then cells were fixed with 250 μL 4%lük (w/v) of paraformaldehyde, permeablized with 250 μL 0.1% (v/v) of Triton X-100 in PBS, and 250 μL 1% (w/v) of BSA in PBST at room temperature (10 min intervals). Then, Phalloidin-iFluor 488 reagent as primary antibody for intracytoplasmic dying performed for staining cytoskeleton of 3T3-L1 adipocytes was prepared in 1:100 (v/v) in 1% (w/v) PBST for 30 min in dark. The After all, of the intracytoplasmic stained cells were observed and imaged at 5X under confocal microscope (Carl-Zeiss LSM 710 NLO, GmbH, Germany).

## Scratch assay

Scratch assay was performed to evaluate the effect of *B. spergulifolius* extracts on the cell migration ability of 3T3-L1 adipocytes [31]. The 3T3-L1 adipocytes were seeded in 6-well plates ($3x10^5$ cells/well). When the cells approximately reached confluence, the scratch-wound in the middle of each well was created with a 200 μL sterile micropipette tip. Then, wells were

removed and washed with DPBS. Medium having $IC_{50}$ dose of *B. spergulifolius* extracts was added to each well. Preadipocytes and adipocytes as controls received, respectively, only fresh medium and adipocyte maintenance medium. All cells were incubated under standard conditions. Images of the scratch-wound were taken at 0. h, 6. h, 24. h, 30. h, 48. h using a microscope (4X magnification). Also, the scratch-wound area was measured from the digital images and calculated using ImageJ software (imagej.nih.gov) [32]. The scratch closure rate (%) was calculated:

$$\text{The scratch closure rate (\%)} = [(\text{Initial scratch-wound width} - \text{Final scratch-wound width})/(\text{Initial scratch-wound width})] \times 100 \quad (3)$$

## Giemsa staining

Giemsa staining was conducted on 3T3-L1 cells to assess the effect of *B. spergulifolius* extracts on morphology. The cells were seeded into a 6-well plate ($3\times10^5$ cells/well) and then confluent cells were treated with plant extracts. After medium was removed and wells were washed with 750 μL PBS twice. To fix cells were incubated 750 μL 60% (v/v) of cold MeOH in PBS for 60 min. MeOH was discarded and 750 μL 100% of MeOH for 10 min at room temperature was added to each well. Then after cells were washed with 750 μL PBS twice. A Giemsa stain stock solution of 0.1 g was prepared by dissolving in 10 mL MeOH for 30 min in heat bath and filtered with Millex Milipore 0.42 μm (Millex-GP, Sigma Aldrich, USA). The fixed cells were stained with from stock solution 1 mL Giemsa from stock solution for 20 min at room temperature, washed with distile water. Then, air-dried cells were observed and imaged at 10X, under an inverted microscope (Leica, Dmil Led Fluo, Thermo Fisher, Germany) [33]. Also, The ImageJ software (imagej.nih.gov) was used for the measurement of the average cell diameter in each three fields in photographs for three wells.

## SEM

Scanning electron microscope (SEM) was used to evaluate the effect of *B. spergulifolius* extracts on the physical morphology of 3T3-L1 adipocytes. The cells were cultured on coverslips (Nunc Thermanox Coverslips, Thermo Scientific, USA) into a 6-well plate ($3\times10^5$ cells/well) and then confluent cells treated with plant extracts. After 24 h, coverslips were transferred into petri dish and cells were fixed with 2.5% (v/v) glutaraldehyde in PBS for 45 min at room temperature. After that, the cells were washed with PBS twice and dehydrated in a graded series of ethanol (respectively, from 30, 50, 70, 80, 90, and 96%) for 15 min at room temperature. Then, cells were incubated with HMDS for 5 min and air-dried samples were mounted with a thin layer of gold and imaged at 500X using SEM (Supra 40 VP, Carl Zeiss, Oberkochen, Germany).

## Statistical analysis

Statistical Package for Social Science (SPSS) v24.0 version (Inc., Chicago, IL, USA) package program was used for statistical analysis of the data. All experiments and measurements were performed in triplicate. All the continuous variables are expressed as the mean ± standard deviation (SD). The graphics were prepared with the Graph Pad Prism software 8.4.3 version (Graphpad Software Inc., USA). $IC_{50}$ is calculated by Graph Pad Prism software 8.4.3 version. It was checked whether the data that obtained in our experiments are normally distributed or not, and Mann-Whitney U or Student's t-test (two-tailed) was used to compare groups. For a comparison of data between more than two independent groups, One-ANOVA tests followed by Tukey's test were used. P values $< 0.05$ were considered statistically significant.

## Results

### Total phenolic contents (TPC) of *B. spergulifolius's* extracts

The extraction yield and TPC of *B. spergulifolius* crude extracts are summarized in Table 3. The extraction efficiency of *B. spergulifolius* Aqueous extract (49.3 ± 1.05%) was the highest, followed by the extraction efficiency of MeOH extract and EA extract (46.91 ± 1.60%, 23.2 ± 1.88%, respectively) lower.

Total phenolic compounds chelate with metal ions that can free radicals and form lipid peroxidation and contribute to antioxidant activity by scavenging free radicals [34, 35]. In our study, the TPC of *B. spergulifolius* extracts was evaluated as an indicator of anti-oxidant activity. TPC in *B. spergulifolius* extracts was calculated using Folin-Ciocalteu reagent and gallic acid equivalent to a standard curve equation ($R^2$ = 0.9970). The TPC was highest in the *B. spergulifolius* EA extract (52.81 ± 1.57 mg/g dry weight). Subsequently, the total phenolic content was lower in *B. spergulifolius* MeOH extract (48.22 ± 1.98 mg/g dry weight) and lowest in *B. spergulifolius* Aqueous extract (30.34 ± 1.36 mg/g dry weight) (Table 3). Each experiment was performed at least 3 times and data were given as the mean ± SD. When the extracts were compared with each other in terms of TPC content, there was a statistically significant difference ($p < 0.05$).

### The cytotoxic effect of *B. spergulifolius* extracts on 3T3-L1 adipocytes

It is a critical step in evaluating the possible cytotoxicity of plant extracts and their suitability for further applications [21, 23]. The cytotoxic effect of *B. spergulifolius* extracts of MeOH (0–1000 µg/mL), EA (0–2000 µg/mL) and Aqueous (0–1000 µg/mL) on mature 3T3-L1 adipocyte cell line was investigated with using MTT assay. After 24 h of incubation with *B. spergulifolius* extracts, the cell viability results are presented in Fig 1.

Cell viability decreased as the extract concentration increased in all extracts. Cell viability began to decrease significantly at concentrations of *B. spergulifolius* MeOH extract higher than 200 µ/mL compared to control (p = 0.033 for 200 µg/mL and p< 0.001 for higher doses). Cell viability began to decrease significantly at concentrations of *B. spergulifolius* EA extract higher than 500 µg/mL compared to control (p< 0.001 for 500 µg/mL and higher doses). Cell viability started to decrease significantly at concentrations of *B. spergulifolius* Aqueous extract higher than 100 µg/mL compared to control (p = 0.001 for 100 µg/mL and p< 0.001 for higher doses) (Fig 1A). However, more than 90% cell viability was preserved in cells exposed to concentrations of less than 10 µg/mL, 350 µg/mL and 25 µg/mL for *B. spergulifolius* MeOH, EA and Aqueous extract, respectively. Also, final concentrations of 0.1% of DMSO, 0.1% of MeOH and 0.1% EA were not cytotoxic to 3T3-L1 cells.

The extract concentration ($IC_{50}$) that causes 50% decrease in the viability of the cells was observed as 305.7 ± 5.583 µg/mL, 567.4 ± 3.008 µg/mL and 418.3 ± 4.390 µg/mL for MeoH, EA

**Table 3.** Yield of extraction (%) and TPCs of *Bolanthus spergulifolius* extracts from different solvents.

| Extract of *B. spergulifolius* | Yield of extraction | Total phenolic content |
|---|---|---|
| | (%, w/w, dry weight) | (mg GAE/g dry weight) |
| MeOH | 46.91 ± 1.60 | *48.22 ± 1.98 |
| EA | 23.2 ± 1.88 | *52.81 ± 1.57 |
| Aqueous | 49.3 ± 1.05 | 30.34 ± 1.36 |

TPC: Total phenolic content; GAE, gallic acid equivalents; MeOH: Methanol; EA: Ethylacetate; Aq: Aqueous. The yield of extraction expressed as % (w/w). TPC expressed as mg GAE/g dry weight extract. Data presented as mean ± S.D. of three independent measurements for each extract. Statistical analysis was conducted using One-way ANOVA test.

*p< 0.05, statistically different from Aqueous.

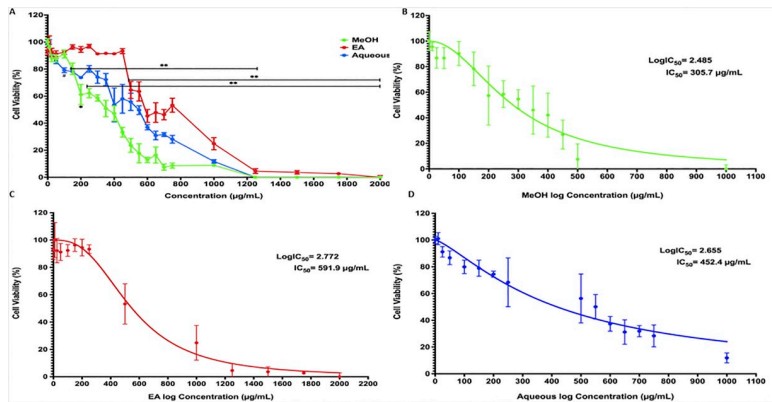

**Fig 1. MTT assay showing the effect $IC_{50}$ value of *B. spergulifolius* extracts on 3T3-L1 cell viability. A.** Effect of *B. spergulifolius* extracts on 3T3-L1 cell viability. The mature 3T3-L1 adipocytes were treated for 24 h with increased concentration of Methanol (MeOH), Ethylacetate (EA), and Aqueous extracts of *B. spergulifolius*. Also, cell treated without crude plant extract was taken as a control. **B-D.** The half-maximal (50%) inhibitory concentration ($IC_{50}$) value of the *B. spergulifolius* extracts. The $IC_{50}$ value is calculated by non-linear regression model. Data expressed as the means ± SD from three independent experiments. To analyse statistically was used one-way ANOVA test. *p = 0.033, #p = 0.001 and **p< 0.001, significantly different from control.

and Aqueous extract, respectively (Fig 1B–1D). Based on these data, the $IC_{50}$ dose of *B. spergulifolius* MeOH, EA and Aqueous extracts was chosen to be applied to the cells in subsequent experiments.

## The effect of *B. spergulifolius* extracts on cell viability in 3T3-L1 adipocytes

A live/dead assay was performed to verify the cytotoxic effect of *B. spergulifolius* extracts on the mature 3T3-L1 adipocyte cell line. Cells were exposed to $IC_{50}$ doses of MeOH, EA and Aqueous of *B. spergulifolius* extracts for 24 h. Then, double staining was performed using C-AM and Eth-1 to distinguish between live (green) and dead (red) cells.

Microscopic images of living and dead cells in preadipocytes, adipocytes and mature 3T3-L1 adipocytes treated for 24 h in staining performed using the live-dead assay were presented in Fig 2A. Under an inverted microscope, intensive green (C-AM) fluorescence was observed in preadipocytes and untreated adipocytes. Red (Eth-1) fluorescence with reduced green (C-AM) fluorescence intensity was observed in mature 3T3-L1 adipocytes treated with *B. spergulifolius* MeOH, EA and Aqueous extracts compared to untreated adipocytes.

The percentage of viable cells after 24 h in preadipocyte and untreated adipocytes was 94.25 ± 2.98% and 97.5 ± 1.29% (p = 0.1836). However, compared to mature 3T3-L1 adipocytes, after 24 h treatment of *B. spergulifolius* MeOH, EA and Aqueous extracts, the percentage of viable cells was 69.75 ± 1.70%, 61.75 ± 1.70%, and 70 ± 4.24% decreased (p< 0.0001, p< 0.0001, and p = 0.0014). Also, compared to Aqueous extract, *B. spergulifolius* MeOH and EA extract treatments resulted in a 0.35% and 11.78% reduction in the percentage of viable cells, respectively (EA-MeOH, EA-Aqueous, and MeOH-Aqueous respectively, p = 0.0122, p = 0.0154, and p = 0.9157) (Fig 2B).

## 3T3-L1 preadipocyte cells differentiated into mature adipocytes and effect of *B. spergulifolius* extracts on lipid accumulation in 3T3-L1 adipocytes

Oil Red O staining was used to confirm the adipogenic phenotype in the differentiation of preadipocytes into mature 3T3-L1 adipocyte cells and to evaluate the effect of *B. spergulifolius* crude extracts on the lipid accumulation ability of mature 3T3-L1 adipocytes [20, 21].

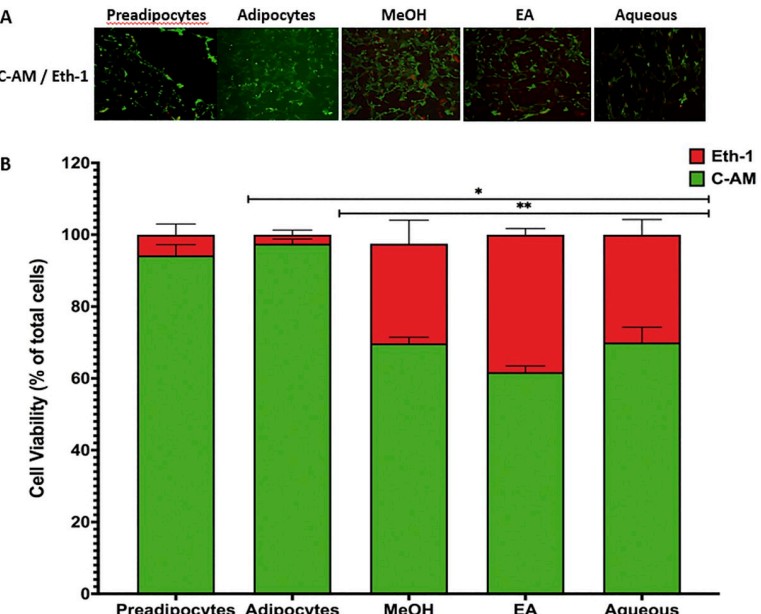

**Fig 2. Live-dead staining showing the effect of *B. spergulifolius* extracts on cell viability in 3T3-L1 adipocytes.** Live-dead staining assay was performed preadipocytes, mature adipocytes without and mature adipocytes with treated with $IC_{50}$ dose of MeOH, EA and Aqueous of *B. spergulifolius* extracts for 24 h **A.** Representative microscopic images at 10X magnification showed 3T3-L1 cells without and treated with *B. spergulifolius* extracts stained with Calcein-Acetomethoxy (C-AM) and Ethidium homodimer-1 (Eth-1). C-AM (green fluorescence) and Eth-1 (red fluorescence) staining showed respectively live and dead 3T3-L1 cells (Scale bar: 100 μm) **B.** Quantitative data of live-dead staining assay. Three coverslips for each cell group were evaluated. Six distinct-randomly chosen regions in each coverslip were visualized and analyzed using microscope and ImageJ software. Data are expressed as a total percentage of viable and dead cells and presented as the means ± SD. To analyze statistical was used Student's t-test (two-tailed). $^{*}p < 0.05$, significantly different from mature 3T3-L1 Adipocytes. $^{**}p < 0.05$, significantly different from Aqueous.

Staining was performed to distinguish lipid droplets in preadipocytes, adipocytes and adipocytes treated with $IC_{50}$ doses of MeOH, EA and Aqueous of *B. spergulifolius* extracts for 24 h with the Oil Red O kit. Under an inverted microscope, more lipid droplets were observed in untreated adipocytes compared to preadipocytes. Lipid accumulation decreased in mature 3T3-L1 adipocytes treated with *B. spergulifolius* MeOH, EA and Aqueous extracts compared to untreated adipocytes (Fig 3A).

After Oil Red O staining performing, levels of lipid accumulation were measured using isopropanol. The $OD_{490}$ value increased by 89.23 ± 6.46% in untreated adipocytes compared to preadipocytes (p < 0.0001). As shown in Fig 3B, compared to untreated mature 3T3-L1 adipocytes, 3T3-L1 adipocytes treated with *B. spergulifolius* MeOH, EA, and Aqueous extract decreased in $OD_{490}$ by respectively, 20.2 ± 2.56%, 23.9 ± 3.16% and 10.3 ± 2.81% (p < 0.0001, p < 0.0001, and p < 0.0016). In addition, $OD_{490}$ values decreased by 11.09% and 15.20%, respectively, in MeOH and EA extract treatment group adipocytes compared to *B. spergulifolius* Aqueous extract (EA-MeOH, EA-Aqueous, and MeOH-Aqueous respectively, p < 0.0572, p < 0.0001, and p < 0.0001) (Fig 3B).

## The effect of *B. spergulifolius* extracts on apoptosis regulated by the *Bax* and *Bcl-2* in 3T3-L1 adipocytes

qRT-PCR was used to evaluate the effect of *B. spergulifolius* extracts on *Bax* and *Bcl-2* gene at mRNA levels associated with apoptosis in mature 3T3-L1 adipocytes [25]. Adipocytes were

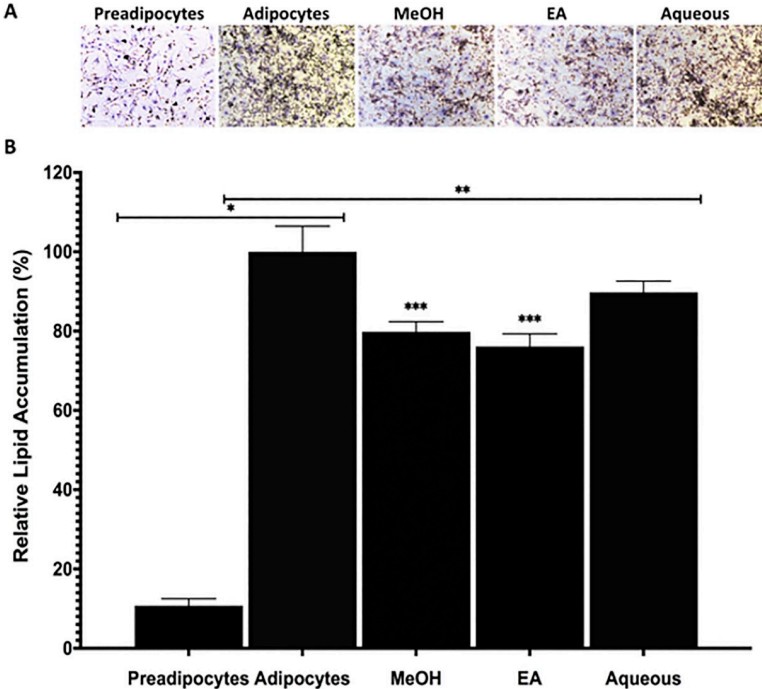

**Fig 3. Oil Red O staining showing 3T3-L1 preadipocyte cells differentiated into mature adipocytes and effects *B. spergulifolius* extracts on lipid accumulation in 3T3-L1 cells. A.** The microscopic images at 10X magnification of lipid droplet of preadipocytes, mature adipocytes without *B. spergulifolius* extracts and adipocytes treated with $IC_{50}$ dose of MeOH, EA and Aqueous of *B. spergulifolius* extracts stained with Oil Red O (Scale bar: 100 μm) **B.** The relative lipid quantization of preadipocytes, mature adipocytes treated without *B. spergulifolius* extracts and with MeOH, EA, Aqueous of *B. spergulifolius extracts*. $OD_{490}$ of Oil Red O staining from three independent experiments was measured. Data are expressed as the relative change of the $OD_{490}$ and presented as the means ± SD. To analyze statistical was used One-way ANOVA test. *$p < 0.0001$ and **$p < 0.001$, significantly different from mature 3T3-L1 Adipocytes. ***$p < 0.001$, significantly different from Aqueous.

treated with $IC_{50}$ doses of MeOH, EA and Aqueous *B. spergulifolius* extracts for 24 h. Compared to preadipocytes, *Bax* and *Bcl-2* genes at mRNA level expression were decreased by 23.22 ± 0.04% and 24.46 ± 1.70% but *Bax/Bcl-2* ratio increased by 1.654 ± 0.01%, respectively in untreated adipocytes (p = 0.0015, p < 0.0001, and p = 0.3451). Pro-apoptotic *Bax* gene at mRNA level expression increased by, respectively, 264.05 ± 0.9%, 347.90 ± 0.57% and 161.64 ± 0.30% in mature 3T3-L1 adipocytes treated with *B. spergulifolius* MeOH, EA and Aqueous extracts compared to untreated adipocytes, while the anti-apoptotic *Bcl-2* gene at mRNA level decreased by 15.71 ± 0.14%, 49.01 ± 0.57%, and 18.13 ± 0.15%, respectively (for *Bax*, p = 0.0020, p = 0.0002, p = 0.0003, and for *Bcl-2* p = 0.0343, p < 0.0001, P = 0.0423, respectively). However, compared to *B. spergulifolius* Aqueous extract, an increase in the expression level of the *Bax* gene was 36.14% and 71.18%, respectively, in MeOH and EA extract treatment groups, while the expression level of the *Bcl-2* gene increased by 2.96% and 37.71%, respectively (MeOH-EA, EA-Aqueous, MeOH-Aqueous, for *Bax* p = 0.0919, p = 0.0039, p = 0.0131 and for *Bcl-2* p = 0.0025, p = 0.0083, p = 0.6994, respectively). Similarly, compared to untreated adipocytes, *B. spergulifolius* MeOH, EA and Aqueous treatment groups mature 3T3-L1 adipocytes showed an increase of 330.80%, 780.70%, and 22.18%, respectively in *Bax/Bcl-2* (p = 0.0057, p = 0.0002 and p = 0.0134, respectively). In addition, compared to Aqueous extract, *B. spergulifolius* MeOH and EA extract treatments increased 33.71% and 173.35% respectively in *Bax/Bcl-2* (EA-MeOH, EA-Aqueous, and MeOH-Aqueous p = 0.0062, p = 0.0025, and p = 0.0936, respectively) (Fig 4).

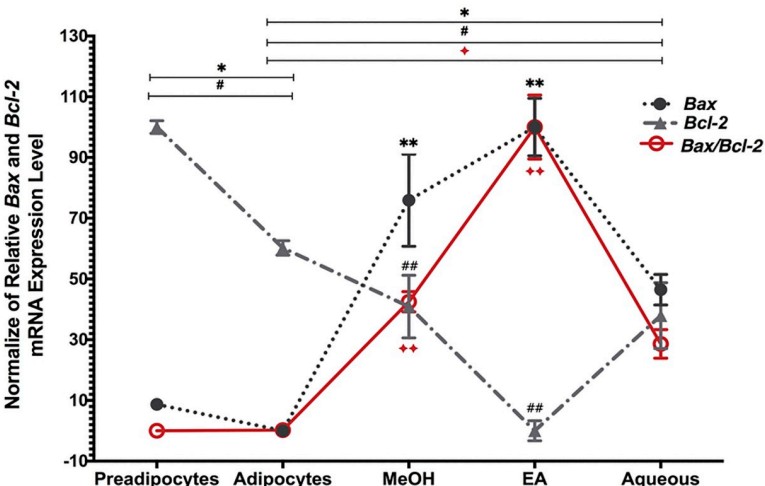

**Fig 4. qRT-PCR data showing the effect of *B. spergulifolius* extracts on apoptosis regulated by the *Bax* and *Bcl-2* in 3T3-L1 adipocytes.** The expression at the mRNA level of *Bcl-2-associated X (Bax)* and *B-cell lymphoma-2 (Bcl-2)* genes and *Bax/Bcl-2* ratio was detected by qRT-PCR in preadipocytes, mature adipocytes without *B. spergulifolius* extracts and adipocytes treated with $IC_{50}$ dose of MeOH, EA and Aqueous of *B. spergulifolius* extracts for 24 h. The *Glyceraldehyde-3-phosphate dehydrogenase* (*Gapdh*) gene was used for normalization. qRT-PCR was analysed using the comparative threshold cycle $C_T$ method ($^{\Delta\Delta}C_T$) from three independent experiments. Data are expressed as the relative change of the $^{\Delta\Delta}C_T$ and the means ± SD. To analyse statistical was used Student's t-test (two-tailed). $^{*}p< 0.05$, $^{\#}p< 0.05$, and $^{\bigstar}p< 0.05$, significantly different from mature 3T3-L1 Adipocytes. $^{**}p< 0.05$, $^{\#\#}p< 0.05$, and $^{\bigstar\bigstar}p< 0.05$, significantly different from Aqueous.

## The effect of *B. spergulifolius* extracts on expression of *Glut-4* in 3T3-L1 adipocytes

qRT-PCR, Western-blot, and immunofluorescence staining were used to evaluate the effect of MeOH, EA, Aqueous of *B. spergulifolius* extracts on *Glut-4* expression in mature 3T3-L1 adipocytes associated with insulin resistance and decreased glucose uptake [26]. *Glut-4* expression was evaluated in untreated preadipocytes, adipocytes and adipocytes treated for 24 h with $IC_{50}$ doses of MeOH, EA and Aqueous of *B. spergulifolius* extracts by qRT-PCR, Western-blot and immunofluorescence staining (Fig 5).

Expression in *Glut-4* mRNA decreased by 389.57 ± 1.60% in untreated adipocytes compared to preadipocytes (p = 0.0004). The mRNA expression level of the *Glut-4* gene increased by 98.84 ± 1.39%, 236.80 ± 0.53%, and 46.18 ± 0.62%, respectively, in mature 3T3-L1 adipocytes treated with *B. spergulifolius* MeOH, EA and Aqueous extracts compared to untreated adipocytes (p = 0.0073, p = 0.0012, p = 0.0458, respectively). However, among the *B. spergulifolius* extract treatment groups, an increase of 36.02% and 130.40% was observed in the mRNA expression level of the *Glut-4* gene in MeOH and EA extract treatment groups, respectively, compared to Aqueous extract (EA-MeOH, EA-Aqueous, and MeOH-Aqueous, for mRNA expression, p = 0.0021, p = 0.0013, p = 0.0106) (Fig 5A).

Western-blot showed a change in the level of Glut-4 protein expression between preadipocyte, adipocyte and treated adipocyte groups (Fig 5B). The relative intensities of Glut-4 protein band in membrane were measured as densitometrically. According to densitometric analysis data expression of Glut-4 protein decreased by 68.93 ± 9.60% in untreated adipocytes compared to preadipocytes (p< 0.0001). Expression of Glut-4 protein levels increased by 26.11 ± 11.65%, 138.66 ± 9.80% and 20.17 ± 7.92%, respectively, in mature 3T3-L1 adipocytes treated with *B. spergulifolius* MeOH, EA and Aqueous extracts compared to untreated adipocytes (p = 0.0291, p< 0.0001, p = 0.004, respectively). However, compared to Aqueous extract,

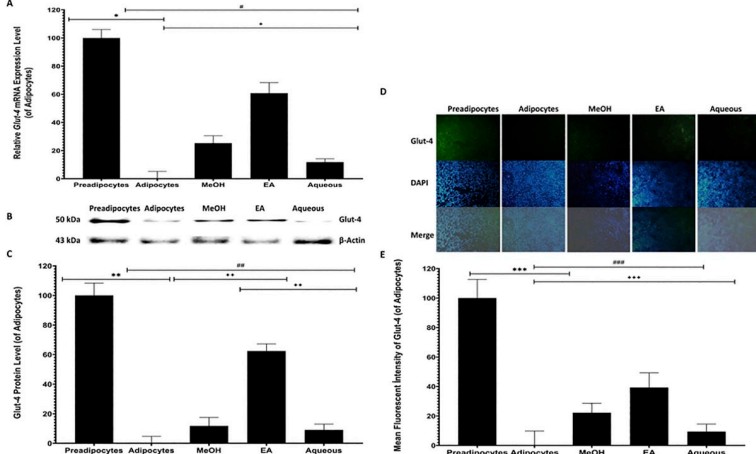

**Fig 5. qRT-PCR, Western-blot, and immunofluorescence staining showing the effect of *B. spergulifolius* extracts on translocation and expression of Glut-4 in 3T3-L1 adipocytes.** The expression of *Glucose transporter-4 (Glut-4)* gene, protein and intra-cellular was quantified by qRT-PCR, Western-blot and immunofluorescence staining in preadipocytes, mature adipocytes without *B. spergulifolius* extracts and adipocytes treated with $IC_{50}$ doses of MeOH, EA and Aqueous of *B. spergulifolius* extracts for 24 h. **A.** *Glut-4* gene expression was detected using qRT-PCR at the mRNA level. *Gapdh* gene was used for normalization. The relative fold change analysis was conducted using the $^{\Delta\Delta}C_T$ method. qRT-PCR was analysed using $^{\Delta\Delta}C_T$ from three independent experiments. Data are expressed as the relative changing of the $^{\Delta\Delta}C_T$ and presented as the means ± SD. **B.** Representative image showing Glut-4 protein (55 kDa) levels in 3T3-L1 cells by Western-blot (20 µg/lane). **C.** The relative intensities of Glut-4 protein band were measured as densitometrically. Beta-Actin (β-Actin) protein was used for normalization. Data were analysed using the ImageLab software. Densitometric analysis data was presented as the relative changing of the band intensities and the means ± SD from three independent experiments. **D.** The microscopic image at 10X magnification showing Glut-4 expression levels and nuclear staining in 3T3-L1 cells by immunofluorescence staining. Merged images obtained using anti-Glut-4 antibody and diamido-2-phenylindole dihydrochloride (DAPI) (Scale bar: 100 µm) **E**. Three coverslips for each cell group were evaluated. Six distinct-randomly chosen regions in each coverslip were pohotographed using a microscope and relative intensities of immunofluorescence staining with anti-Glut4 antibody were quantified using ImageJ software. The immunofluorescence staining analysis data are presented as the relative changing of the fluorescein signal intensities and the means ± SD. To analyze statistical was used Student's t-test (two-tailed). $^*p< 0.05$, $^{**}p< 0.0001$, and $^{***}p< 0.0001$, significantly different from Preadipocytes. $^\#p< 0.05$, $^{\#\#}p< 0.05$ and $^{\#\#\#}p< 0.05$, significantly different from mature 3T3-L1 Adipocytes. $^\blacklozenge p< 0.05$, $^{\blacklozenge\blacklozenge}p< 0.05$, and $^{\blacklozenge\blacklozenge\blacklozenge}p< 0.05$, significantly different from treatment groups.

Glut-4 protein expression levels increased by respectively, 4.93% and 98.59% in the MeOH and EA extract treatment groups (EA-MeOH, EA-Aqueous and MeOH-Aqueous respectively, p< 0.0001, p< 0.0001 and p = 0.3523) (Fig 5C).

Under an inverted microscope, anti-Glut-4 antibody fluorescence was weak in untreated adipocytes compared to preadipocytes. However, anti-Glut-4 antibody fluorescence density increased in mature 3T3-L1 adipocytes treated with *B. spergulifolius* MeOH, EA and Aqueous extracts compared to untreated adipocytes (Fig 5D). According to immunofluorescence staining analysis data, anti-Glut-4 antibody fluorescence intensity decreased 59.87 ± 13.35% in untreated adipocytes compared to preadipocytes (p< 0.0001). Compared to untreated adipocytes, anti-Glut-4 antibody fluorescence density increased by 33.28 ± 8.68%, 58.73 ± 13.46%, and 14.09 ± 6.96%, respectively, in mature 3T3-L1 adipocytes treated with *B. spergulifolius* MeOH, EA and Aqueous extracts (p = 0.0079, p = 0.0021, p = 0.071, respectively). However, anti-Glut-4 antibody fluorescence density in adipocytes increased by respectively, 16.813% and 39.12%, in the MeOH and EA extract treatment groups compared to Aqueous extract (EA-MeOH, EA-Aqueous, and MeOH-Aqueous p = 0.0028, p = 0.0007 and p = 0.0152) (Fig 5E).

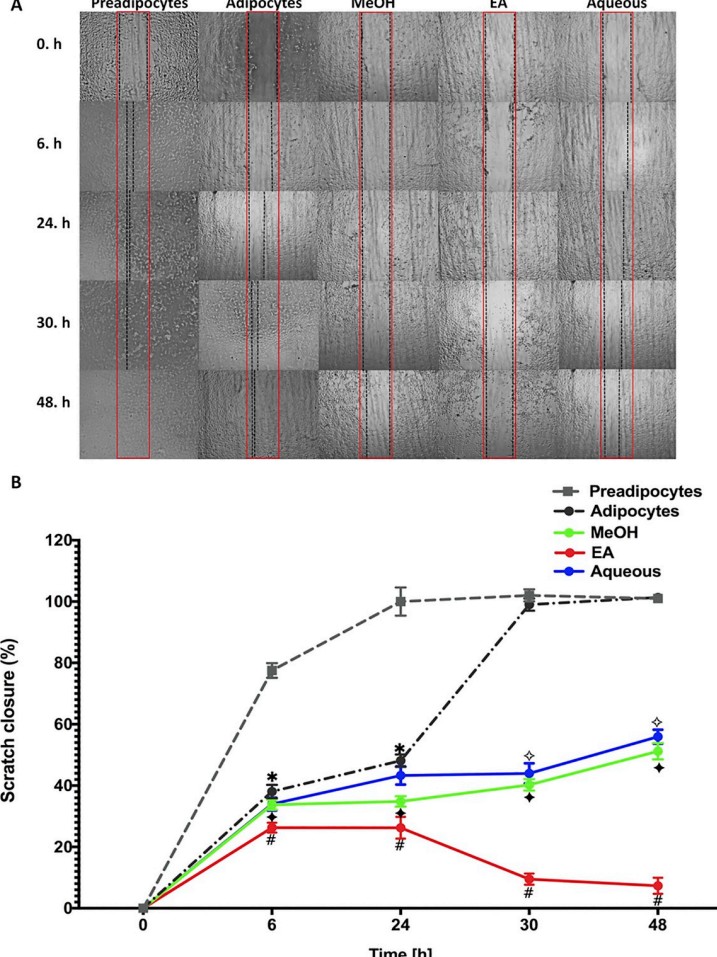

**Fig 6. Scratch assay showing the effect of *B. spergulifolius* extracts on the cell migration ability of 3T3-L1 adipocytes.** The cell migration ability was determined by Scratch assay in preadipocytes, mature adipocytes without *B. spergulifolius* extracts and adipocytes treated with $IC_{50}$ dose of MeOH, EA and Aqueous of *B. spergulifolius* extracts. Cells were cultured to confluent cell monolayers were scratched using the micropipette tip incubated for 0–48 h. **A.** Representative microscopic images (at 10X magnification) captured at 0. h, 6. h, 24. h, 30. h and 48. h showing *in vitro* scratch assay wound healing nature of *B. spergulifolius* extracts. The boundaries of the wounds in cells were presented by the red (initial scratch wound) and black (final scratch wound) colored lines (Scale bar: 100 μm) **B.** The scratch closure rate was quantified using ImageJ software. Data are presented as a percentage of the closure of scratch-wound area at time 0. h and the means ± SD from three independent experiments. To analyze statistical was used one-ANOVA tests followed by Tukey's test were used. *p< 0.05, significantly different from Preadipocytes. ◇p< 0.05, ✦p< 0.05, and #p< 0.05, significantly different from mature 3T3-L1 Adipocytes.

## The effect of *B. spergulifolius* extracts on the cell migration ability of 3T3-L1 adipocytes

*In vitro* scratch assay was used to evaluate the effects of MeOH, EA, Aqueous of *B. spergulifolius* extracts on the migration of mature 3T3-L1 adipocytes [31, 32]. Cell migration and wound closure in untreated preadipocytes, adipocytes and adipocytes treated with $IC_{50}$ doses of MeOH, EA and Aqueous of *B. spergulifolius* extracts for 24 h were evaluated with the Scratch assay. The scratch wound was viewed under the microscope at the 0th, 6th, 24th, 30th and 48th hour (Fig 6).

Under the inverted microscope, a large wound area was observed in untreated adipocytes compared to preadipocytes. Compared to untreated adipocytes, the treatment group mature 3T3-L1 adipocytes treated with the $IC_{50}$ dose of *B. spergulifolius* MeOH, EA and Aqueous extracts significantly inhibited wound healing (Fig 6A).

Scratch-wound area to preadipocytes at $0^{th}$, $6^{th}$, $24^{th}$, $30^{th}$ and $48^{th}$ hour decreased by $77.52 \pm 2.40\%$, $100 \pm 4.58\%$, $100 \pm 2.00\%$, and $100 \pm 1\%$, respectively (Comparing to previous point of time p = 0.0003, p = 0.0234, p = 0.4380, p = 0.6220). Scratch-wound area of adipocytes decreased at $0^{th}$, $6^{th}$, $24^{th}$, $30^{th}$ and $48^{th}$ hour $38.07 \pm 2.17\%$, $48.11 \pm 2.03\%$, $99 \pm 2.00\%$, and $100 \pm 0.05\%$, respectively (Comparing to previous point of time, p = 0.0011, p = 0.0002, p = 0.0002, p = 0.1181). Scratch-wound area of MeOH decreased at $0^{th}$, $6^{th}$, $24^{th}$, $30^{th}$ and $48^{th}$ hour $33.76 \pm 1.63\%$, $34.83 \pm 1.69\%$, $40.24 \pm 1.89\%$, and $51.18 \pm 2.65\%$ respectively (Comparing to a previous point of time, p = 0.0008, p = 0.4590, p = 0.0448, p = 0.0129). Scratch-wound area of EA decreased at $6^{th}$, $24^{th}$, $30^{th}$ and $48^{th}$ hour $26.27 \pm 1.59\%$, $26.25 \pm 3.55\%$, $9.50 \pm 1.83\%$, and $7.33 \pm 2.63\%$ respectively (Comparing to a previous point of time, p = 0.0012, p = 0.9934, p = 0.0035, p = 0.4523). Scratch-wound area of Aqueous decreased at $6^{th}$, $24^{th}$, $30^{th}$ and $48^{th}$ hour $33.96 \pm 2.10\%$, $43.29 \pm 2.96\%$, $43.97 \pm 3.28\%$ and $55.95 \pm 2.25\%$, respectively (Comparing to a previous point of time, p = 0.0013, p = 0.0029, p = 0.3035, p = 0.0209) (Fig 6B).

The percentage of the scratch wound closure was calculated using ImageJ software as the ratio between the scratch-wound area at the initial and at each time point of the assay [32]. According to data analysis of the percentage of the scratch wound closure, scratch-wound area of untreated adipocytes compared to pre-adipocytes decreased at $6^{th}$, $24^{th}$, $30^{th}$ and $48^{th}$ hour $50.88 \pm 2.17\%$, $51.88 \pm 2.03\%$, $2.94 \pm 2.00\%$, and $0.33 \pm 2.65\%$ respectively (p = 0.0015, p = 0.0053, p = 0.3235, p = 0.4226). Compared to untreated adipocytes, in the treatment group mature 3T3-L1 adipocytes treated with *B. spergulifolius* MeOH, EA, and Aqueous extracts, at $6^{th}$ hour scratch-wound area decreased by $11.33 \pm 1.63\%$, $30.99 \pm 1.59\%$, and $10 \pm 2.10\%$, respectively (p = 0.0052, p = 0.0009, p = 0.1061, respectively). In the treatment group $24^{th}$ hour scratch-wound area decreased by $27.60 \pm 1.69\%$, $45.43 \pm 3.55\%$, and $10.01 \pm 2.96\%$ respectively (p = 0.0108, p = 0.0130, p = 0.1167, respectively). $30^{th}$ hour migration area decreased by $59.35 \pm 1.89\%$, $90.40 \pm 1.83\%$, and $55.58 \pm 3.28\%$ respectively (p = 0.0002, p = 0.0002, p = 0.0025, respectively). Also, $48^{th}$ hour scratch-wound area decreased by $49.49 \pm 2.65\%$, $92.75 \pm 2.63\%$, and $44.78 \pm 2.25\%$ respectively (p = 0.0013, p = 0.0003, p = 0.0009, respectively) (Fig 6B).

In addition, scratch-wound area at $6^{th}$ hour decreased by 0.60% and 22.63% respectively, in the MeOH and EA extract treatment groups compared to Aqueous extract (EA-MeOH, EA-Aqueous, and MeOH-Aqueous respectively, p = 0.0002, p = 0.0236 and p = 0.8878). Scratch-wound area at $24^{th}$ hour decreased by 19.54% and 39.36% in the MeOH and EA extract treatment groups compared to Aqueous extract, respectively (EA-MeOH, EA-Aqueous, and MeOH-Aqueous respectively, p = 0.1055, p = 0.0454, and p = 0.0076). Scratch-wound area at $30^{th}$ hour decreased by 8.48% and 78.39% respectively, in the MeOH and EA extract treatment groups compared to Aqueous extract (EA-MeOH, EA-Aqueous, and MeOH-Aqueous respectively, p = 0.0032, p = 0.0072, and p = 0.2028). Scratch-wound area at $48^{th}$ hour decreased by 8.51% and 86.88% respectively, in the MeOH and EA extract treatment groups compared to Aqueous extract (EA-MeOH, EA-Aqueous, and MeOH-Aqueous respectively, p = 0.0004, p< 0.0001 and p = 0.0384) (Fig 6B).

## The effect of *B. spergulifolius* extracts on the cell morphology and diameter of 3T3-L1 adipocytes

Microscopic examination and SEM, Immunofluorescent and Giemsa staining were used to evaluate the effect of MeOH, EA, Aqueous of *B. spergulifolius* extracts on morphology in

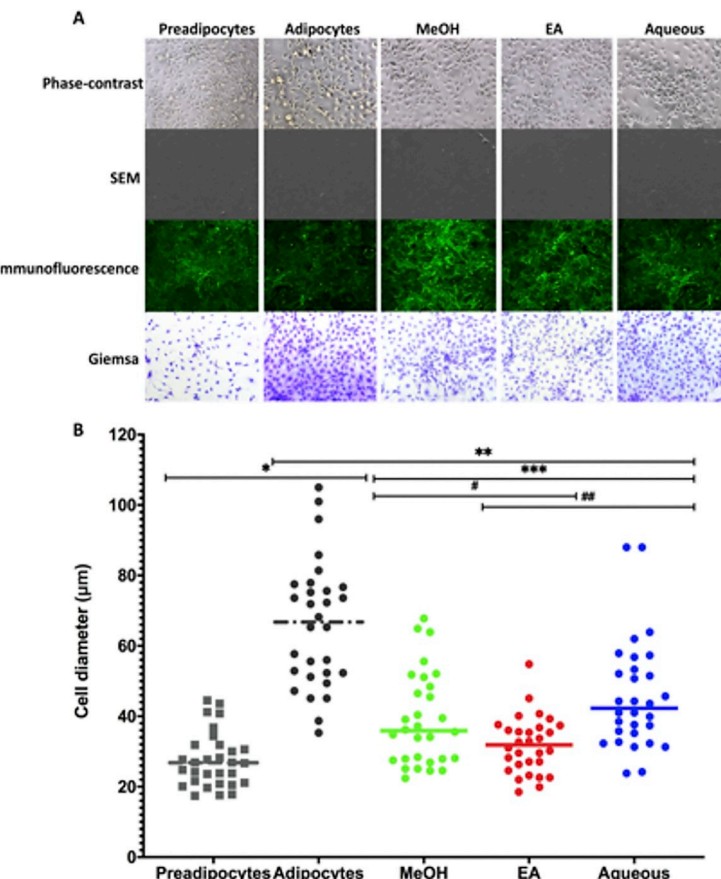

**Fig 7. The phase-contrast microscopy, SEM, Immunofluorescence staining and Giemsa staining showing the effect of *B. spergulifolius* extracts on the cell morphology and diameter of 3T3-L1 adipocytes. A.** Cell morphology was evaluated by phase-contrast microscopy, scanning electron microscopy (SEM), immunofluorescence staining and Giemsa staining in preadipocytes, mature adipocytes without *B. spergulifolius* extracts and adipocytes treated with $IC_{50}$ doses of MeOH, EA and Aqueous of *B. spergulifolius* extracts. Representative images showing changes of cell-morphology after treatment with an $IC_{50}$ dose of MeOH, EA and Aqueous of *B. spergulifolius* extracts for 24 h. The cells were imaged under the phase-contrast microscope at 10X magnification (Scale bar: 100 μm). The cells were mounted with a thin layer of gold and imaged at 500X using scanning electron microscope (SEM) (Scale bar: 200 μm). By immunofluorescence with Phalloidin (Alexa Fluor-488), F-Actin was stained and the cells were viewed under confocal microscopy at 5X magnification (Scale bar: 100 μm). By Giemsa, the cells were stained and viewed under a phase-contrast microscope at 10X magnification (Scale bar: 100 μm). **B.** Measurement of cell diameter in images stained with Giemsa. Three wells for each cell group were evaluated. Three distinct-randomly chosen regions in each well were imaged under a microscope and measurement of the average cell-diameter were quantified using the ImageJ software. Data presented as the average change of the cell diameter and the means ± SD. To analyze statistical was used one-ANOVA tests followed by Tukey's test were used. *p< 0.0001, significantly different from Preadipocytes. **p< 0.0001, significantly different from mature 3T3-L1 Adipocytes. #p = 0.0152 and ##p< 0.0001, significantly different from MeOH and Aqueous.

mature 3T3-L1 adipocytes. Cells in untreated adipocytes showed spherical morphology compared to preadipocytes with phase-contrast microscopy and SEM, immunofluorescent staining and Giemsa staining. No significant morphologic difference was observed in the treatment group mature 3T3-L1 adipocytes treated with $IC_{50}$ doses of *B. spergulifolius* MeOH, EA and Aqueous extracts compared to untreated adipocytes (Fig 7A). However, cells were observed in less rounded morphology in the *B. spergulifolius* EA extract treatment group compared to the other treatment groups.

The ImageJ software was used for the measurement of the average cell diameter in images stained by Giemsa. The cell diameter increased by 139.66 ± 12.94% in untreated 3T3-L1

adipocytes compared to preadipocytes ($p< 0.0001$). Cell diameter decreased by $41.10 \pm 4.8\%$, $51.74 \pm 6.37\%$, and $30.65 \pm 7.60\%$, respectively, in the treatment group mature 3T3-L1 adipocytes treated with *B. spergulifolius* MeOH, EA and Aqueous extracts compared to untreated adipocytes ($p< 0.0001$, $p< 0.0001$, and $p< 0.0001$, respectively). In addition, cell diameter decreased 15.06% and 30.40%, respectively, in MEOH and EA extracts compared to Aqueous extract (EA-MeOH, EA-Aqueous, and MeOH-Aqueous respectively, $p = 0.0152$, $p< 0.0001$, and $p = 0.0695$) (Fig 7B).

## Discussion

DM is a global public health problem, unaware that one in two people in the world has diabetes [1, 2]. T2DM is the most common etiological type of DM, and it puts a huge burden on healthcare and economy, especially in developed and developing countries [36]. It is a complex, chronic disease involving dysfunctions in carbohydrate, fat, protein metabolism and/or in the endocrine system in target cell-tissues, including reduced daily activity, obesity, adipocyte, pancreatic β cells. Hypoglycemia and glycemic fluctuations in T2DM predispose to the development of pathologies including eye, kidney and neural structures, especially with cardiovascular-diseases [1]. In addition, adipocyte hypertrophy is characterized by the induction of adipogenesis and lipogenesis by the change in adiponectin and adipokine synthesis. Increased energy demand in adipocytes causes fat accumulation, oxygen demand, insufficient mitochondrial count despite low-grade local inflammation, and the development of resistance to insulin, leading to the emergence of the T2DM phenotype [4]. However, current antidiabetics show various side effects such as hypoglycemia, weight gain and gastrointestinal problems [5, 6]. Therefore, the discovery and production of safe, natural phyto-pharmaceutical and/or nutraceutical compounds that will target the molecular pathogenesis of T2DM represent new strategies that will contribute to the reduction of the underlying risk factors, prevention and/or treatment of disease.

Caryophyllaceae species are used by various ethnic groups for medical purposes in diseases such as cough, eye and skin problems, dysentery, headache, heart diseases, diabetes, kidney stones, cataracts, epilepsy, asthma, fever, hypertension and menstrual irregularities [8]. *Bolanthus (Ser.) Reichb.*, the Caryophyllaceae family, which includes its genus, has been reported to be the natural source of biologically active compounds for phyto-pharmaceuticals [12]. Therefore, in our study, potential anti-diabetic effects of crude extracts obtained from *B. spergulifolius* with three solvents on mature 3T3-L1 adipocytes were investigated.

Phenolic compounds are powerful scavengers of hydroxyl and peroxyl radicals due to their (-OH) group content, they are metal binders and affect stopping the Fenton and Haber-Weiss reactions that are the source of reactive oxygen species (ROS) [35]. It has been reported that phenolic compounds taken in the diet play a crucial role in reducing oxidative stress and protecting against diabetes, cardiovascular diseases and cancer [37]. However, it has been reported that phenolic compounds contribute to glycemic control and can be used for treating DM [3, 38]. Therefore, in our study, first, *B. spergulifolius* was extracted with different solvents. Extraction is a critical stage in obtaining phytochemicals from the plant. The efficiency of the extraction performed; It affects many parameters such as phytochemicals, method used, solvent, etc. [39]. In our study, extracts of *B. spergulifolius* were obtained using MeOH, EA and Aqueous, and the extraction efficiency was observed as 46.91%, 23.2%, and 49.3%, respectively. The extraction efficiency increased in relation to the polarity of the solvent used in the extraction [40]. This may be due to the higher solubility of biomacromolecules in Aqueous and MeOH than EA. Many assays such as TPC, DPPH scavenging assay, ABTS scavenging assay and Ferric reducing antioxidant power (FRAP) are used to investigate the antioxidant capacity of

herbal extracts. In our study, the total phenolic content of *B. spergulifolius* extracts was investigated by Folin-Caiocalteu to evaluate the extracellular antioxidant activity [18]. TPC values for MeOH, EA, and Aqueous were observed as 48.22 mg GAE/g, 52.81 mg GAE/g and 30.34 mg GAE/g *B. spergulifolius*, respectively. The lower TPC value in Aqueous extract compared to other solvents may be related to the higher non-phenolic compound content. However, the high content of soluble phenolic compounds in MeOH and EA TPC value can be explained by the fact that they form complex, contain phenolic-containing phenolics and/or these compounds have high molecular weights. Although the Caryophyllaceae family is rich in total phenolic content, the literature on the TPC value of *B. spergulifolius* is limited [17]. Our results showed that *B. spergulifoliu*s extracts had a significant phenolic content and MeOH and EA were the better extraction solvent for *B. spergulifolius*.

The MTT test based on the mitochondrial metabolic capacity of living cells was used to evaluate the potential cytotoxicity of *B. spergulifolius* extracts on mature 3T3-L1 adipocyte cells and the suitability of the extracts for further investigation [23]. After 24 h incubation with *B. spergulifolius* MeOH, EA and Aqueous extracts, cell viability decreased with an increase in extract concentration >200 μg/mL, ≥500 μg/mL, >100 μg/mL, respectively. However, cells exposed to extract concentrations of 10 μg/mL, 450 μg/mL, and ≤25 μg/mL, respectively, maintained over 90% cell viability. $IC_{50}$ values for *B. spergulifolius* MeOH, EA and Aqueous extracts were observed as 305.7 μg/mL, 567.4 μg/mL, and 418.3 μg/mL, respectively. These doses were used in further applications to investigate the potential anti-diabetic effects of the extracts on mature 3T3-L1 adipocytes. MTT data showed that *B. spergulifolius* MEOH extract has higher cytotoxic potential than EA and Aqueous extracts. In addition, to confirm the MTT assay data in our study, the cytotoxic activity of the $IC_{50}$ dose of the extracts on mature 3T3-L1 adipocytes was evaluated using the live-dead assay [24]. Cell viability was observed to be 69.75%, 61.75%, and 70% for *B. spergulifolius* MeOH, EA and Aqueous extracts, respectively. These results confirmed that the extracts of *B. spergulifolius* examined were potential cytotoxic agents for the cell line studied.

Cell lines as in vitro model systems have been provided opportunity to understand effects of plant extracts on animal and human health for a long time. The 3T3-L1 preadipocyte cell line have a fibroblast-like morphology can differentiate into the adipocyte phenotype under favorable conditions, and is a useful *in vitro* model that is widely used to study adipogenesis *in vitro* [19, 20, 21]. When 3T3-L1 preadipocytes in the post-confluent $G_0$ phase are continuously exposed to an adipogenic factor regimen with various agents, including insulin, they undergo a period of mitotic clonal expansion and then undergo differentiation and growth ceases. The ability of this cell line to differentiate into mature adipocytes is well characterized. The process of adipogenesis includes alteration of cell morphology, growth arrest and clonal expansion, and lipid storage [19]. In adipocyte differentiation, mitochondrial activity becomes important and requires a large number of mitochondria in the process of maturation. Matured and large adipocytes develop resistance to insulin due to the insufficient number of mitochondria. The insufficient number of mitochondria and lipid accumulation is caused an increasead of ROT level [41]. The 3T3-L1 cell line was selected for the current study because it demonstrates relevant characteristics including lipid accumulation and glucose homeostasis. Lipid accumulation in the cytoplasm of mature 3T3-L1 adipocytes can be easily observed with Oil Red O staining [42]. In our study, it was shown that pre-adipocytes successfully differentiated into 3T3-L1 mature adipocytes after 14 days. In addition, in our study, the effect of *B. spergulifolius* extracts on lipid accumulation in mature 3T3-L1 adipocytes was evaluated with Oil Red O staining. *B. spergulifolius* MeOH reduced the lipid accumulation of EA and Aqueous extracts by 20.2%, 23.9% and 10.3%, respectively. Our data show that *B. spergulifolius* extracts can be used to inhibit lipid accumulation in mature 3T3-L1 adipocytes.

Transcriptional activity of some major genes involved in the insulin signaling pathway, such as *Peroxisome proliferator-activated receptor gamma (Ppary)* and *Glucose transporter-4 (Glut-4)* changes during the maturation of adipocytes to promote the uptake of glucose and fatty acids in adipogenesis [43, 44]. Glut-4 is responsible for glucose uptake in muscle and adipose tissue by an insulin-regulated mechanism [45]. In response to insulin, *Glut-4* expression increases and the translocation of Glut-4 containing vesicles to the plasma membrane is induced, resulting in glucose uptake into the cell [46]. In T2DM, the expression of *Glut-4*, which is insufficiently stimulated due to insulin resistance, decreases, its localization is impaired and glucose transport in target tissues is prevented [3, 47]. Since insulin resistance and decreased Glut-4 levels in adipocyte tissue are characteristic, the development of insulin-sensitizing agents against T2DM is an important current research topic [26, 48]. Studies have reported that in differentiated human adipocyte and *in vitro* mature 3T3-L1 cells, the expression and translocation of the *Glut-4* gene at the mRNA, protein level decreased [42, 49]. In our study, qRT-PCR, western-blot, and immunofluorescence staining data were consistent with each other, showing that the treatment of mature 3T3-L1 adipocytes with *B. spergulifolius* raw extracts caused a significant increase in expression of the *Glut-4* gene at the mRNA, protein and intracellular level. Considering the critical importance of Glut-4's activity in adipose tissue and its positive correlation with insulin sensitivity, our data show that *B. spergulifolius* can be used in reducing insulin-resistance in mature adipocytes. Moreover, the results demonstrated that *B. spergulifolius* MeOH and EA extracts had high and so close TPC value compare to Aqueous, but its EA extract possesses the ability to promote of Glut-4 expression in the 3T3-L1 adipocytes more than other extracts. On this basis, it may not be a linear correlation between total phenolic content of the extracts and ability to promote Glut-4 expression of the extracts. From the data, it suggested that it was associated with individual type of phenolic compound in TPC or/and content of bioactive compounds such as polyphenols, flavonoids, tannins, alkaloids, glycosides, saponins which have different chemical characteristics and polarities [17, 39, 50]. Therefore, the bioactive compounds in *B. spergulifolius* should be invategated, identified and quantification with further studies.

Although apoptosis in adipocytes is not fully understood, adipocytes are highly apoptosis-resistant. Studies have reported that adipogenesis in 3T3-L1 adipocytes can be inhibited by apoptosis induction. In this way, it has been demonstrated that adipose tissue mass can be reduced by induction of apoptosis [51, 52]. The induction of apoptosis in mature adipocytes can be a critical strategy for treating many metabolic diseases such as diabetes and obesity. Therefore, we investigated whether *B. spergulifolius* extracts had apoptotic activity in mature adipocytes. The balance between Bax/Bcl-2 and caspase-3 activation plays a key role in apoptosis [25]. In addition, several natural compounds such as capsaicin, xanthohumol and isoxanthohumol, resveratrol and quercetin induce apoptosis in 3T3-L1 adipocytes through up-regulation of *Bax* and down-regulation of *Bcl-2* in mature adipocytes [53–55]. In our study, pro-apoptotic *Bax* genes of *B. spergulifolius* MeOH, EA and Aqueous extracts in mature 3T3-L1 adipocytes was 264.05%, 347.90%, and 161.64%, respectively, and the anti-apoptotic *Bcl-2* gene was 15.71%, 49.01%, respectively and down-regulated 18.13%. Based on the relationship between the change in *Bax/Bcl-2* ratio and apoptosis, our result shows that *B. spergulifolius* induces apoptosis in mature adipocytes. Furthermore, in the present study, when 3T3-L1 adipocytes were treated with *B. supergliofolius* extracts with the incresing doses, their viabilities appeared to be reduced but apoptotic status not evaluated in a dose-dependent-manner. Also, about half of 3T3-L1 adipocytes lost their viabilities and this value induced strongly apoptosis in 3T3-L1 adipocytes by treating with $IC_{50}$ value of *B. spergulifolius* extracts in 24 h. However, in further studies it is necessary to investigate the dose as well as time-dependent effects of *B. spergulifolius* MeOH, EA and Aqueous extracts on the induction of apoptosis in 3T3-L1 adipocytes.

*In vitro* and *in vivo* models are available to evaluate the wound healing nature of new therapeutic agents. Of these, the scratch assay is a practical, inexpensive and well-established *in vitro* model that provides the first evidence of cell migration and wound healing efficacy of potential therapeutic agents. Cell proliferation and migration are essential components of the wound healing process. In the Scratch assay, scratches are created *in vitro*, causing the monolayer cell layer to deteriorate and the loss of cell-cell contact. Cell migration and proliferation are encouraged by the induction of various factors in the wound area over time [31, 32]. Therefore, in our study, the scratch assay was used to evaluate the effect of *B. spergulifolius* extracts on cell migration and wound healing capacity in mature adipocytes. Migration and wound healing in the cells were observed for up to 48 h. At $30^{th}$ hour, complete wound healing was observed in the wells containing untreated preadipocytes and adipocytes. However, when *B. spergulifolius* extracts were treated with $IC_{50}$ value, cell migration was slowed in adipocytes and wound healing was delayed. The extracts of *B. spergulifolius* MeOH, EA and Aqueous inhibited migration and healing by 59.35%, 90.40%, and 55.58%, respectively, in the wound area, especially after 30 h. These data show that *B. spergulifolius* delayed wound healing in mature 3T3-L1 adipocytes. This may be related to the suppression of adipogenesis, proliferation and induction of apoptosis in adipocytes [56].

3T3-L1 preadipocytes are morphologically similar to fibroblasts. The induction of differentiation is directly related to increased globular cell morphology, glucose-uptake and lipid accumulation. Adipogenesis and lipogenesis contribute to adipocyte hypertrophy by inducing lipid accumulation, reorganization of the cytoskeleton, loss of fibroblastic morphology, resulting in a spherical shape, and increase in cell size [57]. Therefore, in our study, we evaluated the effect of *B. spergulifolius* extracts on cell morphology and cell-diameter in mature adipocytes using phase-contrast, SEM, immunofluorescence and Giemsa staining. Preadipocytes lost their fibroblastic morphology because of differentiation from differentiation and acquired a spheric form, which is the morphological feature of mature adipocytes. However, in a mature 3T3-L1 adipocytes, extracts of *B. spergulifolius* MeOH, EA and Aqueous had a negative effect on the diameter of the cells and suppressed 41.10%, 51.74%, and 30.65%, respectively. Our data, *B. spergulifolius* may reduce lipid accumulation in mature adipocytes, contributing to the reduction of the cell diameter and gaining fibroblastic morphology.

The current study has some potential limits. One of them is the study were performed only at the *in vitro* level. In order to determine the antidiabetic role of *B. spergulifolius* more clearly and to understand the underlying mechanisms better, it is necessary to investigate the related pathways in the molecular pathogenesis of diabetes. Advanced multidisciplinary studies at the *in vivo* level will confirm the antidiabetic effect of *B. spergulifolius*, our *in vitro* data, and provide new data that will lead to its use in the clinic. Furthermore, insulin to stimulate glucose uptake into muscle and adipose tissue is essential for normal glucose homeostasis. In differentiation process, 3T3-L1 preadipocytes matured adipocytes with about 20-fold induced in the number of insulin receptors and acquire the ability to uptake glucose in reply to insulin [58]. However, the impact of plant extracts on glucose uptake into 3T3-L1 adipocytes was not examined in the present study. On the other hand, the encouraging outcomes of B. supergulifolius extracts could be attributed to the total phenolic content, but main polyphenols and other bioactive molecules in the extracts need to be identified by biochemical and The High-Performance Liquid chromatography (HPLC)-based assays [59].

## Conclusion

Our study reveals the *in vitro* bioactive capacity of the plant by showing that *B. spergulifolius* extracts reduce lipid accumulation, delay wound healing, induce apoptosis and increase Glut-4

expression. These preliminary data can also provide a basis for developing phyto-pharmaceutical strategies aimed at the discovery of components to be isolated from *B. spergulifolius* and the acquisition of products. All of our results show that *B. spergulifolius* can improve its functional capacity in the future and thus play an important role in the prevention and/or treatment of T2DM. Pharmacological, phytochemical properties of this plant extract and biochemical effects on some signaling pathways of ıts active compounds should be further investigated as a natural medicine for diabetes treatment.

## Supporting information

**S1 Raw image. Glut-4 protein levels in 3T3-L1 cells by Western-blot.**
(DOCX)

## Author Contributions

**Conceptualization:** Sibel Özdaş, Murat Koç.

**Data curation:** Gizem Ece Derici, Sibel Özdaş, İpek Canatar.

**Formal analysis:** Gizem Ece Derici, Sibel Özdaş, İpek Canatar, Murat Koç.

**Funding acquisition:** Gizem Ece Derici, Sibel Özdaş.

**Investigation:** Sibel Özdaş, İpek Canatar.

**Methodology:** Sibel Özdaş, İpek Canatar, Murat Koç.

**Project administration:** Gizem Ece Derici, Sibel Özdaş.

**Resources:** Gizem Ece Derici, Sibel Özdaş.

**Software:** Sibel Özdaş, İpek Canatar.

**Supervision:** Sibel Özdaş.

**Validation:** Sibel Özdaş, Murat Koç.

**Visualization:** Gizem Ece Derici, Sibel Özdaş, İpek Canatar, Murat Koç.

**Writing – original draft:** Gizem Ece Derici, Sibel Özdaş, İpek Canatar, Murat Koç.

**Writing – review & editing:** Gizem Ece Derici, Sibel Özdaş, İpek Canatar.

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
