## [Decision Letter · Decision Letter 0]

16 Mar 2021

PONE-D-21-04167

Antidiabetic activities of Bolanthus spergulifolius (Caryophyllaceae) extracts on insulin-resistant 3T3-L1 adipocytes

PLOS ONE

Dear Dr. Özdaş,

Thank you for submitting your manuscript to PLOS ONE. After careful consideration, we feel that it has merit but does not fully meet PLOS ONE’s publication criteria as it currently stands. Therefore, we invite you to submit a revised version of the manuscript that addresses the points raised during the review process.

We look forward to receiving your revised manuscript.

Kind regards,

Yi Cao

Academic Editor

PLOS ONE

Journal Requirements:

3. In your Methods section, please provide additional location information of the collection site, including geographic coordinates for the data set if available.

4. In your Methods section, please provide additional information regarding the permits you obtained for the work. Please ensure you have included the full name of the authority that approved the collection site access and, if no permits were required, a brief statement explaining why.

7. Thank you for stating the following in the Acknowledgments Section of your manuscript:

„This study was financially supported by The Adana Alparslan Türkeş Science and Technology University Scientific Research Projects Coordination Unit (Project No: 19332006 and Project No: 20103005). The authors wish to thank the Laboratory of Nanotechnology Research and Application (Project number: TR62/18/ÜRET/0032) for infrastructure support.”

„The author(s) received no specific funding for this work.”

Reviewers' comments:

Reviewer's Responses to Questions

**Comments to the Author**

1. Is the manuscript technically sound, and do the data support the conclusions?

Reviewer #1: Yes

Reviewer #2: Yes

2. Has the statistical analysis been performed appropriately and rigorously? 

Reviewer #1: Yes

Reviewer #2: Yes

3. Have the authors made all data underlying the findings in their manuscript fully available?

Reviewer #1: Yes

Reviewer #2: Yes

4. Is the manuscript presented in an intelligible fashion and written in standard English?

Reviewer #1: Yes

Reviewer #2: Yes

5. Review Comments to the Author

Reviewer #1: Abstract

1. Please remove effect in the second line.

2. Rephrase the second paragraph as: The total phenolic contents of methanolic, ethyl acettate and aqueous extracts of B. spergulifolius were evaluated via Folin-Ciocateau.

3. What is MTT? Provide the full meaning

4. Redefine your IC50

5. Change P<0.05 to p<0.05

6. Remove (SEM0

Introduction

1. Italize in vitro

2. Clearly state the problem and justification of your research.

3. Some of the references are is square bracket [ ] while some are in ( ), please, be consistent.

Materials and Methods

1. Add reagents to chemicals, that is, Chemicals and reagents

2. There is no need of including the plant picture, since it is aresearch article

3. Please, represent all your tables using the standard method

Results

1. What is TPCs, please, write in full

Conclusion

This section needs to be represented. summarize your findings and profer further study.

Reviewer #2: The authors investigated the anti-diabetic effects in vitro of B. spergulifolius extracts on 3T3-L1 adipocyte. The assays were well designed and performed. The results and discussions were also persuasive. However, some issues should be addressed before a possible publication. The details are as follows:

1\\The abstract is too long！！

2\\The resolution of the figures is poor.

3\\Page 17 line 6:“100” to “100%” & “X” to” ×”

4\\Page 17 line 18:“100” to “100%” & “X” to” ×”

5\\There is no provided information on why the authors chose 3T3-L1 adipocytes as model cell.

6\\The effects of the extracts on glucose uptake in 3T3-L1 preadipocytes should be added.

7\\Since the total phenolic contents of MeOH and EA are so close, how do we understand the significant difference of the effects on Glt-4 protein expressions between them.

8\\ Since the used dose of the extract induced apoptosis in mature adipocytes, the authors should consider the influences of the doses on the activities.

9\\The main polyphenols in the extracts are highly suggested to be identified by HPLS-MS.

6. PLOS authors have the option to publish the peer review history of their article (what does this mean?). If published, this will include your full peer review and any attached files.

Reviewer #1: **Yes: **OLADEJI, Oluwole Solomon

Reviewer #2: No

---

## [Author Response · Author response to Decision Letter 0]

2 May 2021

Dear Editor, 

Yi Cao 

PLOS ONE

We revised our manuscript PONE-D-21-04167 entitled “Antidiabetic activities of Bolanthus spergulifolius (Caryophyllaceae) extracts on insulin-resistant 3T3-L1 adipocytes” in the light of the comments of the reviewers. All changes made in the text have been marked with yellow hightligth. I hope our manuscript now meets the standards of your esteemed journal.

Sincerely yours,

Sibel ÖZDAŞ, PhD 

Adana Alparslan Türkeş Science and Technology University, 

Faculty of Engineering, Department of Bioengineering, Adana, Turkey

Tel: +90 505 5118947

E-mail: sozdas@atu.edu.tr

Journal Requirements: When submitting your revision, we need you to address these additional requirements.

Requirement 1. Please ensure that your manuscript meets PLOS ONE's style requirements, including those for file naming. The PLOS ONE style templates can be found at

Answer 1. We thank you for those valuable comments. We ensured that our manuscript meets PLOS ONE's style requirements and corrected our paper style.

Requirement 2. Please review your reference list to ensure that it is complete and correct. If you have cited papers that have been retracted, please include the rationale for doing so in the manuscript text, or remove these references and replace them with relevant current references. Any changes to the reference list should be mentioned in the rebuttal letter that accompanies your revised manuscript. If you need to cite a retracted article, indicate the article’s retracted status in the References list and also include a citation and full reference for the retraction notice.

Answer 2. We reviewed and ensured that our current reference list meets PLOS ONE’S style requirements.

Requirement 3. In your Methods section, please provide additional location information of the collection site, including geographic coordinates for the data set if available.

Answer 3. We have given the geographical coordinates of the plant location in the plant material section of the method section. We added the following sentences to the Plant material in Methods (1rd paragraph): “B. spergulifolius used in this study was collected from Kütahya province, Gediz district, Murat Mountain, 1495 m, Kaplıcalar road, 38°56'48'' N - 29°36'44'' E, serpentine soils, grassy levels in Turkey during the flowering season in 2019. Identification and authentication of the plant material were conducted by Dr. Murat Koç, taxonomist, from Department of Traditional, Complementary and Integrative Medicine Graduate School of Public Health, Ankara Yıldırım Beyazıt University, Turkey. A voucher specimen of B. spergulifolius (No. Koç2043&Hamzaoğlu) were deposited in the herbarium (Biology Department of Bozok University, Turkey) [11]. The air-dried plant materials under shade were ground in a coffee grinder (Krups 75, France) [17]. 

Requirement 4. In your Methods section, please provide additional information regarding the permits you obtained for the work. Please ensure you have included the full name of the authority that approved the collection site access and, if no permits were required, a brief statement explaining why. 

Answer 4. No permit has been obtained from the institutions for the collection of the plants. Because there is no requirement to obtain permits in national projects. 

Requirement 5. PLOS ONE now requires that authors provide the original uncropped and unadjusted images underlying all blot or gel results reported in a submission’s figures or Supporting Information files. This policy and the journal’s other requirements for blot/gel reporting and figure preparation are described in detail at https://journals.plos.org/plosone/s/figures#loc-blot-and-gel-reporting-requirements and https://journals.plos.org/plosone/s/figures#loc-preparing-figures-from-image-files. When you submit your revised manuscript, please ensure that your figures adhere fully to these guidelines and provide the original underlying images for all blot or gel data reported in your submission. See the following link for instructions on providing the original image data: https://journals.plos.org/plosone/s/figures#loc-original-images-for-blots-and-gels.

Answer 5: We will provide the original uncropped and unadjusted image underlying Fig. 5.B Western-blot result reported in a Supporting Information files. The URLs/accession numbers will be available only after acceptance of the manuscript for publication so that we can ensure their inclusion before publication

Requirement 6. In your Data Availability statement, you have not specified where the minimal data set underlying the results described in your manuscript can be found. PLOS defines a study's minimal data set as the underlying data used to reach the conclusions drawn in the manuscript and any additional data required to replicate the reported study findings in their entirety. All PLOS journals require that the minimal data set be made fully available. For more information about our data policy, please see http://journals.plos.org/plosone/s/data-availability.

Answer 6. In our Data Availability statement, we had not specified where the minimal data set. We know requirement that our minimal data set be made fully available. We were look into your data policy. Therefore, we changed our data availability statement as followings: “yes all data are fully available without restriction”. However, we cannot provide accession numbers or DOIs for my data set at submission. We ticked, “the URLs/accession numbers/DOIs will be available only after acceptance of the manuscript for publication so that we can ensure their inclusion before publication” in submission system

Requirement 7. Thank you for stating the following in the Acknowledgments Section of your manuscript: 

“This study was financially supported by The Adana Alparslan Türkeş Science and Technology University Scientific Research Projects Coordination Unit (Project No: 19332006 and Project No: 20103005). The authors wish to thank the Laboratory of Nanotechnology Research and Application (Project number: TR62/18/ÜRET/0032) for infrastructure support.”

Answer 7. The funding statement has been removed from the Acknowledgements section of manuscript and necessary arrangements have been made in the online submission form.

We updated our statement in Acknowledgments the as follows: “The researchers would like to thank the collaboration Laboratory of Nanotechnology Research and Application (TR62/18/ÜRET/0032) that has made this work possible.” 

Our Funding Statement corrected in online submission system as follows: “Adana Science and Technology University Scientific Research Projects Coordination Unit: Award Number: Project No: 19332006 and Project No: 20103005 | Recipient: Sibel ÖZDAŞ, PhD”

Reviewers' comments:

Reviewer's Responses to Questions Comments to the Author

1. Is the manuscript technically sound, and do the data support the conclusions?

Reviewer #1: Yes Reviewer #2: Yes 

 2. Has the statistical analysis been performed appropriately and rigorously?

Reviewer #1: Yes Reviewer #2: Yes 

 3. Have the authors made all data underlying the findings in their manuscript fully available?

Reviewer #1: Yes Reviewer #2: Yes 4. Is the manuscript presented in an intelligible fashion and written in standard English?

Reviewer #1: Yes Reviewer #2: Yes 

5. Review Comments to the Author

Reviewer #1:

Reviewer #1, Question 1A: A. Abstract

Reviewer #1, Question 1A.1: Please remove effect in the second line.

Reviewer #1, Answer 1A.1: We thank you for those valuable comments. We removed effect in the second line.

Reviewer #1, Question 1A.2: Rephrase the second paragraph as: The total phenolic contents of methanolic, ethyl acettate and aqueous extracts of B. spergulifolius were evaluated via Folin-Ciocateau.

Reviewer #1, Answer 1A.2: We rephrased the second paragraph in Abstract as follows: “...In this study, it was explored the potential anti-diabetic effects in vitro of B. spergulifolius extracts on 3T3-L1 adipocytes. The total phenolic contents (TPC) of methanolic (MeOH), ethyl acettate (EA) and aqueous extracts of B. spergulifolius were evaluated via Folin-Ciocateau. B. spergulifolius extracts showing highly TPC (Aqueous< MeOH< EA) and their different concentrations were carried out on preadipocytes differentiated in to mature 3T3-L1 adipocytes to investigate their half-maximal (50%) inhibitory concentration (IC50) value by using Thiazolyl blue tetrazolium bromide (MTT) assay. The IC50 of MeOH, EA and Aqueous extracts were observed as 305.7 ± 5.583 µg/mL, 567.4 ± 3.008 µg/mL, and 418.3 ± 4.390 µg/mL and used for further experiments...”

Reviewer #1, Question 1A.3: What is MTT? Provide the full meaning

Reviewer #1, Answer 1A.3: The full meaning of MTT provided as (Tris, 3-(4,5-dimethylthiazol-2-yl)-2,5-diphenyltetrazolium bromide) in the relevant part of the abstract section. We rewirettin Abstract as follows: “...B. spergulifolius extracts showing highly TPC (Aqueous< MeOH< EA) and their different concentrations were carried out on preadipocytes differentiated in to mature 3T3-L1 adipocytes to investigate their half-maximal (50%) inhibitory concentration (IC50) value by using Thiazolyl blue tetrazolium bromide (MTT) assay...”

Reviewer #1, Question 1A.4: Redefine your IC50

Reviewer #1, Answer 1A.4: We redefined IC50 value as “half-maximal (50%) inhibitory concentration (IC50) value” in the relevant part of the abstract section.

Reviewer #1, Question 1A.5: 5. Change P<0.05 to p<0.05

Reviewer #1, Answer 1A.5: We changed p values in the whole paper from “P” to “p”.

Reviewer #1, Question 1A.6: Remove (SEM)

Reviewer #1, Answer 1A.6: We removed (SEM) in the relevant part of the abstract section.

Reviewer #1, Question 1B: B. Introduction

Reviewer #1, Question 1B.1: Italize in vitro

Reviewer #1, Answer 1B.1: We italized “in vitro” as “in vitro” in the relevant part of the introduction section.

Reviewer #1, Question 1B.2: Clearly state the problem and justification of your research.

Reviewer #1, Answer 1B.2: We clearly expressed the problem and changed the last paragraph of Introduction Section as follows: “Bolanthus (Ser.) Reichb., is one of the genera of the family Caryophyllaceae. The aspect that makes this genus important for Turkey is that the entire taxa are endemic to Turkey [11, 14, 15]. Bolanthus spergulifolius (Jaub. & Spach) Hub.-Mor. (B. spergulifolius) was first defined with the name Heterochroa spergulifolia by Jaubert and Spach in 1843 [16]. Later on, the taxon was transferred to the genus Bolanthus by Huber-Morath in 1967 [15]. Cayophyllaceae family has been traditionally used in therapeutic medicine for some deseases [8]. Some studies have shown that Cayophyllaceae family has compounds such as saponins, flavonoids, phenolic acids, phytoecdysteroids with strong bioactive properties and can be resources in the research of new drug active ingredients [12]. However, biological activities of B. spergulifolius are poorly described yet. Therefore, in this study, it was aimed to investigate the anti-diabetic effects in vitro of B. spergulifolius extracts on 3T3-L1 adipocytes.”.

Reviewer #1, Question 1B.3: Some of the references are is square bracket [ ] while some are in ( ), please, be consistent.

Reviewer #1, Answer 1B.3: We corrected all reference citation as in square bracket [ ] .

Reviewer #1, Question 1C: C. Materials and Methods

Reviewer #1, Question 1C.1: Add reagents to chemicals, that is, Chemicals and reagents

Reviewer #1, Answer 1C.1: We changed title as “Chemicals and reagents” in Materials and Method Section.

Reviewer #1, Question 1C.2: There is no need of including the plant picture, since it is aresearch article

Reviewer #1, Answer 1C.2: We removed plant picture in Figure 1.

Reviewer #1, Question 1C.3: Please, represent all your tables using the standard method

Reviewer #1, Answer 1C.3: We standardized all of our tables.

Reviewer #1, Question 1D: D. Results

Reviewer #1, Question 1D.1: What is TPCs, please, write in full

Reviewer #1, Answer 1D.1: We changed title as “Total Phenolic Contents (TPC) of B. spergulifolius’s extracts”

Reviewer #1, Question 1E: E.Conclusion

Reviewer #1, Question 1E.1:This section needs to be represented. summarize your findings and profer further study.

Reviewer #1, Answer 1E.1: We summarized our findings and profered further study. We rewirettin Conclusion section as follows: “.....Our study reveals the in vitro bioactive capacity of the plant by showing that B. spergulifolius extracts reduce lipid accumulation, delay wound healing, induce apoptosis and increase Glut-4 expression. These preliminary data can also provide a basis for developing phyto-pharmaceutical strategies aimed at the discovery of components to be isolated from B. spergulifolius and the acquisition of products. All of our results show that B. spergulifolius can improve its functional capacity in the future and thus play an important role in the prevention and/or treatment of T2DM. Pharmacological, phytochemical properties of this plant extract and biochemical effects on some signaling pathways of ıts active compounds should be further investigated as a natural medicine for diabetes treatment.

Reviewer #2: The authors investigated the anti-diabetic effects in vitro of B. spergulifolius extracts on 3T3-L1 adipocyte. The assays were well designed and performed. The results and discussions were also persuasive. However, some issues should be addressed before a possible publication. The details are as follows:

Reviewer #2, Question 1: The abstract is too long！

Reviewer #2, Answer 1: We thank you for those valuable comments. We arranged Abstract section according to your comment as: “Diabetes mellitus (DM) is a metabolic disorder with chronic hyperglycemia featured by metabolic outcomes owing to insufficient insulin secretion and/or insulin effect defect. It is critical to investigate new therapeutic approaches for T2DM and alternative, natural agents that target molecules in potential signal pathways. Medicinal plants are significant resources in the research of alternative new drug active ingredients. Bolanthus spergulifolius (B. spergulifolius) is one of the genera of the family Caryophyllaceae. In this study, it was explored the potential anti-diabetic effects in vitro of B. spergulifolius extracts on 3T3-L1 adipocytes. The total phenolic contents (TPC) of methanolic (MeOH), ethyl acettate (EA) and aqueous extracts of B. spergulifolius were evaluated via Folin-Ciocateau. B. spergulifolius extracts showing highly TPC (Aqueous< MeOH< EA) and their different concentrations were carried out on preadipocytes differentiated in to mature 3T3-L1 adipocytes to investigate their half-maximal (50%) inhibitory concentration (IC50) value by using Thiazolyl blue tetrazolium bromide (MTT) assay. The IC50 of MeOH, EA and Aqueous extracts were observed as 305.7 ± 5.583 µg/mL, 567.4 ± 3.008 µg/mL, and 418.3 ± 4.390 µg/mL and used for further experiments. A live/dead assay further confirmed the cytotoxic effects of MeOH, EA and Aqueous extracts (respectively, 69.75 ± 1.70%, 61.75 ± 1.70%, 70 ± 4.24%, and for all p< 0.05). Also, effects of extracts on lipid accumulation in mature 3T3-L1 adipocytes were evaluated by Oil-Red O staining assay. The extracts effectively decreased lipid-accumulation compared to untreated adipocytes (for all p< 0.05). Moreover, effect of extracts on apoptosis regulated by the Bax and Bcl-2 was investigated by quantitative reverse transcription polymerase chain reaction (qRT-PCR). The extracts significantly induced apoptosis by up-regulating pro-apoptotic Bax expression but down-regulated anti-apoptotic Bcl-2 gene expression compared to untreated adipocytes (for all p< 0.05). The Glut-4 expression linked with insulin resistance was determined by qRT-PCR, Western-blot analysis, and immunofluorescence staining. In parallel, the expression of Glut-4 in adipocytes treated with extracts was significantly higher compared to untreated adipocytes (for all p< 0.05). Extracts significantly suppressed cell migration after 30 h of wounding in a scratch-assay (for all p< 0.05). Cell morphology and diameter were further evaluated by phase-contrast microscopy, scanning electron microscopy, Immunofluorescence with F-Actin and Giemsa staining. The adipocytes treated with extracts partially lost spherical morphology and showed smaller cell-diameter compared to untreated adipocytes (for all p< 0.05). In conclusion, these results suggest that extracts of B. spergulifolius cause to an induce apoptosis, decrease lipid-accumulation, wound healing, up-regulating Glut-4 level and might contribute to reducing of insulin-resistance in DM.”

Reviewer #2, Question 2: 2\\The resolution of the figures is poor.

Reviewer #2, Answer 2: We improved resolution of the figures.

Reviewer #2, Question 3: 3\\Page 17 line 6:“100” to “100%” & “X” to” ×”

Reviewer #2, Answer 3: We corrected in Page 17 line 6

Reviewer #2, Question 4: 4\\Page 17 line 18:“100” to “100%” & “X” to” ×”

Reviewer #2, Answer 4: We corrected in Page 17 line 18

Reviewer #2, Question 5: 5\\There is no provided information on why the authors chose 3T3-L1 adipocytes as model cell.

Reviewer #2, Answer 5: We added information on why we used 3T3-L1 adipocytes as model cell in this study. We rewirettin Discussion Section (5rd paragraph) as follows: “Cell lines as in vitro model systems have been provided opportunity to understand effects of plant extracts on animal and human health for a long time. The 3T3-L1 preadipocyte cell line have a fibroblast-like morphology can differentiate into the adipocyte phenotype under favorable conditions, and is a useful in vitro model that is widely used to study adipogenesis in vitro [19, 20, 21]. When 3T3-L1 preadipocytes in the post-confluent G0 phase are continuously exposed to an adipogenic factor regimen with various agents, including insulin, they undergo a period of mitotic clonal expansion and then undergo differentiation and growth ceases. The ability of this cell line to differentiate into mature adipocytes is well characterized. The process of adipogenesis includes alteration of cell morphology, growth arrest and clonal expansion, and lipid storage [19]. In adipocyte differentiation, mitochondrial activity becomes important and requires a large number of mitochondria in the process of maturation. Matured and large adipocytes develop resistance to insulin due to the insufficient number of mitochondria. The insufficient number of mitochondria and lipid accumulation is caused an incresead of ROT level [41]. The 3T3-L1 cell line was selected for the current study because it demonstrates relevant characteristics including lipid accumulation and glucose homeostasis...”

Reviewer #2, Question 6: 6\\The effects of the extracts on glucose uptake in 3T3-L1 preadipocytes should be added.

Reviewer #2, Answer 6: Our study has significant financial limits. The glucose uptake assay could not be analyzed due to the financial limitations of our study. So, we rewirettin Discussion Section (10rd paragraph) as follows: “The current study has some potantial limits. One of them is the study were performed only at the in vitro level. In order to determine the antidiabetic role of B. spergulifolius more clearly and to understand the underlying mechanisms better, it is necessary to investigate the related pathways in the molecular pathogenesis of diabetes. Advanced multidisciplinary studies at the in vivo level will confirm the antidiabetic effect of B. spergulifolius, our in vitro data, and provide new data that will lead to its use in the clinic. Furthermore, insulin to stimulate glucose uptake into muscle and adipose tissue is essential for normal glucose homeostasis. In differentiation process, 3T3-L1 preadipocytes matured adipocytes with about 20-fold induced in the number of insulin receptors and acquire the ability to uptake glucose in reply to insulin [57]. However, the impact of plant extracts on glucose uptake into 3T3-L1 adipocytes was not examined in the present study...”

Reviewer #2, Question 7: 7\\Since the total phenolic contents of MeOH and EA are so close, how do we understand the significant difference of the effects on Glt-4 protein expressions between them.

Reviewer #2, Answer 7: We rewirettin Discussion Section (6rd paragraph) as follows: “...Moreover, the results demonstrated that B. spergulifolius MeOH and EA extracts had high and so close TPC value compare to Aqueous, but ıts EA extract possesses the ability to promote of Glut-4 expression in the 3T3-L1 adipocytes more than other extracts. On this basis, it may not be a linear correlation between total phenolic content of the extracts and ability to promote Glut-4 expression of the extracts. From the data, it suggested that it was associated with individual type of phenolic compound in TPC or/and content of bioactive compounds such as polyphenols, flavonoids, tannins, alkaloids, glycosides, saponins which have different chemical characteristics and polarities [17, 39, 50]. Therefore, the bioactive compounds in B. spergulifolius should be invategated, identified and quantification with further studies.”

Reviewer #2, Question 8: 8\\ Since the used dose of the extract induced apoptosis in mature adipocytes, the authors should consider the influences of the doses on the activities.

Reviewer #2, Answer 8: According to your comment we rewirettin Discussion Section (10rd paragraph) as follows: “In our study, pro-apoptotic Bax genes of B. spergulifolius MeOH, EA and Aqueous extracts in mature 3T3-L1 adipocytes was 264.05%, 347.90%, and 161.64%, respectively, and the anti-apoptotic Bcl-2 gene was 15.71%, 49.01%, respectively and down-regulated 18.13%. Based on the relationship between the change in Bax/Bcl-2 ratio and apoptosis, our result shows that B. spergulifolius induces apoptosis in mature adipocytes. Furthermore, in the present study, when 3T3-L1 adipocytes were treated with B. supergliofolius extracts with the incresing doses, their viabilities appeared to be reduced but apoptotic status not evaluated in a dose-dependent-manner. Also, about half of 3T3-L1 adipocytes lost their viabilities and this value induced strongly apoptosis in 3T3-L1 adipocytes by treating with IC50 value of B. spergulifolius extracts in 24 h. However, in further studies it is necessary to investigate the dose as well as time-dependent effects of B. spergulifolius MeOH, EA and Aqueous extracts on the induction of apoptosis in 3T3-L1 adipocytes.”

Reviewer #2, Question 9: 9\\The main polyphenols in the extracts are highly suggested to be identified by HPLS-MS.

Reviewer #2, Answer 9: The High Performance Liquid chromatography (HPLC) is one of most common analytical technique used in pharmaceutical industry for determination and quantification of drug substances and its related substances [57]. Our study has significant financial limits. HPLC, HPLC-MS methods could not be analyzed due to the financial limitations of our study. Therofore, only the total phenolic content of B. spergulifolius was considered within the potential limits of our study. According to your comment we rewirettin Discussion Section (10rd paragraph) as follows: “The current study has some potantial limits. One of them is the study were performed only at the in vitro level. In order to determine the antidiabetic role of B. spergulifolius more clearly and to understand the underlying mechanisms better, it is necessary to investigate the related pathways in the molecular pathogenesis of diabetes. Advanced multidisciplinary studies at the in vivo level will confirm the antidiabetic effect of B. spergulifolius, our in vitro data, and provide new data that will lead to its use in the clinic. Furthermore, insulin to stimulate glucose uptake into muscle and adipose tissue is essential for normal glucose homeostasis. In differentiation process, 3T3-L1 preadipocytes matured adipocytes with about 20-fold induced in the number of insulin receptors and acquire the ability to uptake glucose in reply to insulin [57]. However, the impact of plant extracts on glucose uptake into 3T3-L1 adipocytes was not examined in the present study. On the other hand, the encouraging outcomes of B. supergulifolius extracts could be attributed to the total phenolic content, but main polyphenols and other bioactive molecules in the extracts need to be identified by biochemical and The High-Performance Liquid chromatography (HPLC)-based assays [58].”

 6. PLOS authors have the option to publish the peer review history of their article (what does this mean?). If published, this will include your full peer review and any attached files.

Do you want your identity to be public for this peer review? For information about this choice, including consent withdrawal, please see our Privacy Policy.

Reviewer #1: Yes: OLADEJI, Oluwole Solomon

Reviewer #2: No

---

## [Decision Letter · Decision Letter 1]

21 May 2021

Antidiabetic activities of Bolanthus spergulifolius (Caryophyllaceae) extracts on insulin-resistant 3T3-L1 adipocytes

PONE-D-21-04167R1

Dear Dr. Özdaş,

We’re pleased to inform you that your manuscript has been judged scientifically suitable for publication and will be formally accepted for publication once it meets all outstanding technical requirements.

Kind regards,

Yi Cao

Academic Editor

PLOS ONE

Additional Editor Comments (optional):

Reviewers' comments:

Reviewer's Responses to Questions

**Comments to the Author**

1. If the authors have adequately addressed your comments raised in a previous round of review and you feel that this manuscript is now acceptable for publication, you may indicate that here to bypass the “Comments to the Author” section, enter your conflict of interest statement in the “Confidential to Editor” section, and submit your "Accept" recommendation.

Reviewer #1: All comments have been addressed

Reviewer #2: All comments have been addressed

2. Is the manuscript technically sound, and do the data support the conclusions?

Reviewer #1: Yes

Reviewer #2: Yes

3. Has the statistical analysis been performed appropriately and rigorously? 

Reviewer #1: (No Response)

Reviewer #2: Yes

4. Have the authors made all data underlying the findings in their manuscript fully available?

Reviewer #1: Yes

Reviewer #2: Yes

5. Is the manuscript presented in an intelligible fashion and written in standard English?

Reviewer #1: Yes

Reviewer #2: Yes

6. Review Comments to the Author

Reviewer #1: The authors have addressed the points and comments made in the previous review. I am glad to recommend this manuscript for acceptance and publication.

Reviewer #2: |The authors have reviesed their manuscript carefully according to the suggestions from the reviews ,accept!

7. PLOS authors have the option to publish the peer review history of their article (what does this mean?). If published, this will include your full peer review and any attached files.

Reviewer #1: No

Reviewer #2: No

---

## [Editor Report · Acceptance letter]

8 Jun 2021

PONE-D-21-04167R1 

Antidiabetic activities of *Bolanthus spergulifolius* (Caryophyllaceae) extracts on insulin-resistant 3T3-L1 adipocytes 

Dear Dr. Özdaş:

I'm pleased to inform you that your manuscript has been deemed suitable for publication in PLOS ONE. Congratulations! Your manuscript is now with our production department. 

Kind regards, 

on behalf of

Dr. Yi Cao 

Academic Editor

PLOS ONE